



**Phenomenology of summer ozone episodes over the Madrid**
**Metropolitan Area, central Spain**
Xavier Querol X.[1], Andrés Alastuey A.[1], Gotzon Gangoiti[2], Noemí Perez[1], Hong K. Lee[3], Heeram
R. Eun[3], Yonghee Park[3], Enrique Mantilla E.[4], Miguel Escudero[5], Gloria Titos[1], Lucio Alonso[2],
Brice Temime-Roussel[6], Nicolas Marchand[6], Juan R. Moreta, M. Arantxa Revuelta[7], Pedro
Salvador [8], Begoña Artíñano[8], Saúl García dos Santos[9], Mónica Anguas[10], Alberto Notario[11],
Alfonso Saiz-Lopez[10], Roy M. Harrison[12], Kang-Ho Ahn[3]
[1]Institute of Environmental Assessment and Water Research (IDAEA-CSIC), C/Jordi Girona 18-26, Barcelona, 08034
Spain
[2]Escuela Técnica Superior Ingeniería de Bilbao, Departamento Ingeniería Química y del Medio Ambiente,
Universidad del País Vasco UPV/EHU, Urkixo Zumarkalea, S/N, Bilbao, 48013  Spain
[3]Department of Mechanical Engineering, Hanyang University, Ansan 425-791, Republic of Korea
[4]Centro de Estudios Ambientales del Mediterráneo, CEAM, Unidad Asociada al CSIC, Parque Tecnológico C/ Charles
R. Darwin, 14 Paterna, Valencia, 46980 Spain
[5] Centro Universitario de la Defensa de Zaragoza, Academia General Militar, Ctra. de Huesca s/n, Zaragoza, 50090
Spain
[6]Aix Marseille Univ, CNRS, LCE, Marseille, France
[7]Agencia Estatal de Meteorología, AEMET, C/ Leonardo Prieto Castro, 8, Madrid, 28071 Spain
[8]Department of Environment, CIEMAT, Joint Research Unit Atmospheric Pollution CIEMAT-CSIC, c/ Avenida
Complutense 40, Madrid, 28040 Spain
[9]Centro Nacional de Sanidad Ambiental. Instituto de Salud Carlos III (ISCIII), Ctr Majadahoda a Pozuelo km 2,
Majadahonda (Madrid), 28222 Spain
[10]Department of Atmospheric Chemistry and Climate, Institute of Physical Chemistry Rocasolano, CSIC, Madrid,
28006 Spain
[11]University of Castilla-La Mancha, Physical Chemistry Department, Faculty of Chemical Science and Technologies,
Ciudad Real, Spain.
[12]National Centre for Atmospheric Science, University of Birmingham, B15 2TT United Kingdom. [+]Also at:
Department of Environmental Sciences/Centre for Excellence in Environmental Studies, King Abdulaziz University,
Jeddah, Saudi Arabia

## 32 Abstract

Various studies have reported that photochemical nucleation of new ultrafine particles (UFP)
in urban environments within high insolation regions occurs simultaneously with high ozone
($O_3$). In this work, we evaluate the atmospheric dynamics leading to summer $O_3$ episodes in the
Madrid Air Basin (Central Iberia) by means of measuring a 3D distribution of concentrations for
both pollutants. To this end, we obtained vertical profiles (up to 1200 m, above ground level)
using tethered balloons and miniaturised instrumentation at a suburban site located to the SW
of the Madrid Metropolitan Area (MMA), Majadahonda site (MJDH) in July 2016.
Simultaneously, measurements of an extensive number of air quality and meteorological
parameters were carried out at 3 supersites across the MMA. Furthermore, data from $O_3$-
soundings and daily radio-sounding were also used to interpret the atmospheric dynamics.
The results demonstrate the concatenation of venting and accumulation episodes, with
relative $O_3$ lows (venting) and peaks (accumulation) in surface levels. Regardless of the episode
type, fumigation of high altitude $O_3$-rich layers contributes the major proportion of surface $O_3$
concentrations. Accumulation episodes are characterised by a relatively thinner planetary
boundary layer (PBL< 1500 m at midday, lower in altitude than the orographic features), low
synoptic winds and the development of mountain breezes along the slope of the Guadarrama
Mountain Range (W and NW of MMA, maximum altitude >2400 m). This orographic-



meteorological setting causes the vertical recirculation of air masses and the enrichment of $O_3$
in the lower tropospheric layers. When the highly polluted urban plume from Madrid is
affected by these dynamics, the highest $O_x$ ($O_3$+$NO_2$) concentrations are recorded in the MMA.
Vertical $O_3$ profiles during venting episodes, with marked synoptic winds and a deepening of
the PBL reaching >2000 m above sea level, were characterised by an upward gradient in $O_3$
levels, whereas low-altitude $O_3$ concentration maxima due to local/regional production were
found during the accumulation episodes. The two contributions to $O_3$ surface levels
(fumigation from high altitude strata and local/regional production) require very different
approaches for policy actions. In contrast to $O_3$ vertical top-down transfer, UFP are formed in
the lowest levels and are transferred upwards progressively with the growth of the PBL.

**Keywords:** Ozone, ultrafine particles, photochemical pollution, air quality, vertical profiles.

**1.  Introduction**
The EU Directive 2008/50/EC, amended by Directive 2015/1480/EC, on ambient air quality
establishes the need to comply with air quality standards to protect citizens and ecosystems. If
these are not met, plans to improve air quality must be implemented. Despite the
considerable improvements in air quality during the last decade, non-compliances with the
European air quality standards are still reported in most Europe. In particular the limit values
for nitrogen dioxide ($NO_2$), particulate matter (PM10 and PM2.5) and tropospheric ozone ($O_3$)
target value are frequently exceeded. Therefore, in 2013, the National Plan for Air Quality and
Protection of the Atmosphere (Plan AIRE) 2013-2016, was drawn up, and approved by the
Council of Ministers' Agreement of 12/04/2013.
Measures to effectively reduce $NO_2$ and PM pollution are relatively easy to identify. However,
defining policies for abating $O_3$, other photochemical pollutants and the secondary
components of PM is much more complex.
Photochemical pollution is a subject of great environmental importance in Southern (S) Europe
due to its climatic and geographical characteristics. Sub-products of this type of contamination
are many, noteworthy tropospheric $O_3$, secondary PM (nitrate, sulphate and secondary organic
compounds), and the generation of new ultra-fine particles (UFP) by nucleation (Gomez-
Moreno et al., 2011, Brines et al., 2015).
The abatement of tropospheric $O_3$ levels in this region is a difficult challenge due to its origin,
which may be local, regional or transboundary (Millán et al., 2000 and Millán, 2014), the
complexity of the meteorological scenarios leading to severe episodes (Millán et al., 1997,
Gangoiti et al., 2001, Dieguez et al., 2009 and 2014 and Millán, 2014), as well as the complexity
of the chemical processes that drive its formation and sinks, which are not linear in many cases
(Monks et al., 2014, and references therein).
This complex context has led to a lack of 'sufficient' $O_3$ abatement in Spain and Europe; while
for primary pollutants, such as $SO_2$ and CO, and the primary fractions of PM10 and PM2.5 the
improvement has been very evident (EEA, 2016). Thus, the latest air quality assessment for





Europe (EEA, 2016) shows that: i) there has been a tendency for the peak $O_3$ concentration
values to decrease in the recent years, but not sufficiently to meet WHO guidelines and EC
standards; and ii) the problem of $O_3$ episodes is more pronounced in the S than in Northern (N)
and Central Europe. Likewise, $O_3$ levels are higher in rural than in urban areas, both due to the
generation process, which requires time from the emission of urban, industrial and biogenic
precursors, and the consumption (NO titration) of $O_3$ that takes place in urban areas. Apart
from this EEA report, other recent studies such as EMEP (2016), Escudero et al., (2014), Garcia
et al (2014) and Querol et al. (2014 and 2016) also evidenced that there is a general tendency
for $O_3$ to increase in urban areas, including traffic sites, probably due to the greater relative
reduction of NO emissions compared to $NO_2$, and therefore to the lower NO titration effect. It
is also found that $O_3$ levels in the regional background have remained constant over the last 15
years, but acute episodes have been drastically reduced compared to the late 1990s, and these
markedly increase during heat waves such as those in summer 2003 and 2015 (EEA, 2016,
Diéguez et al., 2009 and 2014 and Querol et al., 2016). A recent study reported that an
increase of 30-40% in ambient air $O_3$ levels along with a decrease of 20-40% in $NO_2$ from 2007
to 2014 in Madrid, may have led to a large concentration increase of up to 70% and 90% in OH
and $NO_3$ (the main tropospheric oxidants), respectively, thereby changing the oxidative
capacity of this urban atmosphere (Saiz-Lopez et al., 2017). We still do not know if this increase
is due to a decrease in the effect of NO titration or to the fact that the $O_3$ formation is by
volatile organic compounds (VOCs) dominated.
Intensive research on $O_3$ pollution has been carried out since the late 1980s in the Western
Mediterranean, which has been key to understand the behaviour of this pollutant in Europe,
and to establish the current air quality European standards (Millán et al., 1991, 1996a, 1996b,
1996c, 2000, 2002; Millán, 2002; EC, 2002, 2004; Millán and Sanz, 1999; Mantilla et al., 1997;
Salvador et al., 1997, 1999; Gangoiti et al., 2001; Stein et al., 2004, 2005; Doval et al., 2012;
Castell et al., 2008a, 2008b, 2012; Millán et al., 2014, Escudero et al., 2014). Diéguez et al.
(2009 and 2014) described in detail the temporal and spatial variation of $O_3$ levels in Spain.
These studies highlight the low inter-annual variability in regional background stations, as well
as the existence of specific areas, such as the Madrid air basin, Northern valleys influenced by
the Barcelona urban plume, Puertollano basin or the interior of the Valencian region, where
very high $O_3$ episodes are relatively frequent, and point to urban and industrial hot spots as
relevant sources of precursors. Recently, Querol et al. (2016) evidenced that the highest $O_3$
episodes, with hourly exceedances of the information threshold to the population (180 µg/m$^3$)
for 2000-2015 occurred mostly around these densely populated or industrialised areas.
Querol et al. (2017) reported that the load of $O_3$ and precursors from the plume of the
metropolitan area of Barcelona contributed decisively to the exceedances of the information
threshold in the northern areas of Barcelona during the acute $O_3$ episodes in July 2015. They
also demonstrated that the meteorology associated was very complex, similar to the scenarios
reported by Gangoiti et al. (2001), Millán (2014) and Diéguez et al. (2014) for other regions of
the Western Mediterranean. Regional transport of $O_3$ is also very relevant, and that acute $O_3$
episodes, exceeding the information threshold, were caused by a dominant regional
contribution (also with high contributions from local formation recirculated during prior days)
to $O_3$, on top of which an additional smaller local 'fresh' contribution was added. It was also



shown that the vast majority of these exceedances are recorded in the month of July of the
respective years.
In addition to the primary emissions, nucleation or new particle formation (NPF) processes give
rise to relevant contributions to the urban ambient air UFP concentrations, mostly during
photochemical pollution episodes in spring and summer (Brines et al., 2015 and references
therein). Ambient conditions favouring urban NPF are high insolation, low relative humidity,
available $SO_2$ and VOCs, as well as low pre-existing particle surface area (low condensation
sink), common features that enhance new particle formation events (Kulmala et al., 2004;
Kulmala and Kerminen, 2008, Sipilä, et al., 2010, Salma et al., 2016).
In this study, we evaluate the temporal and spatial variability of $O_3$ (and UFP) in the Madrid
city/basin, to investigate the causes of acute summer episodes of both pollutants, and to
investigate possible inter-relationships. In a subsequent twin article we will focus on the
phenomenology of UFP nucleation episodes linked with these photochemical events. Data on
UFP are included in this paper only where they assist in interpreting the behaviour of $O_3$.

## 2.  Methodology

### 2.1. The study area

The Madrid air basin and the Madrid Metropolitan Area (MMA) are located in the central
plain, or Meseta, of the Iberian Peninsula at around 700 m a.s.l. Regarding the topographic
features, the Guadarrama range which runs in the NE-SW direction reaches heights up to 2400
m a.s.l. and is located 40 km north from the MMA. To the S, are the Toledo Mountains which
run from E to W (Figure 1). Lower mountains are also located to the NE and E, which are part
of the Iberian range. Consequently, the Madrid plain shows a NE-SW channelling of winds,
forced by the main mountain ranges, and following the basin of the Tagus River and its
tributaries. In particular, the MMA is located to the NE of the river basin and at its right side.
Climatologically, the area is characterised by continental conditions with hot summers and
cold winters with both seasons typically being dry. Mean annual precipitation of around 400
mm is mainly concentrated in autumn and spring. The MMA is one of the most densely
populated regions in Spain, with more than 5 million inhabitants, including Madrid City and
surrounding towns. According to Salvador et al. (2015), the main anthropogenic emissions are
dominated by road traffic and residential heating (in winter), with minor contributions from
industry and a large airport.
Plaza et al. (1997), Pujadas et al. (2000) and Artíñano et al. (2003) described the major
meteorological patterns affecting the dispersion of pollutants in the basin, and their
seasonality. For summer, Plaza et al. (1997) concluded that the development of strong thermal
convective activity and the influence of the mountain ranges produce characteristic mesoscale
re-circulations and the development of a very deep mixing layer (Crespí et al., 1995). These
authors report that these re-circulations contribute markedly to the high $O_3$ episodes recorded
in the region. According to Plaza et al (1997) and Diéguez et al. (2009 and 2014) the
arrangement of the Guadarrama range favours the early heating of its S slopes that causes a
clockwise turning of wind direction from a NE component during the night, towards an E and S
during the early morning and midday, and to the SW during the late afternoon thus defining



the north-western sector downwind the city as the prone area for $O_3$ transport. Night time
downslope winds inside the basin induce the observed north-easterlies at lower levels.
Influenced by these contributions, the barrier effect of the Guadarrama range against the N
and NW (Atlantic) winds, as well as the repeated clockwise circulation described above, cause
movement of the urban plume of Madrid across the basin. This meteorological system allows
local $O_3$ formation and transport. Regarding the vertical scale, Plaza et al. (1997) also showed
that fumigation from high $O_3$-rich layers (injected by upslope winds the previous day or days,
or transported from other areas outside the Madrid basin) could also contribute to enhance
the surface $O_3$ concentrations across the basin. This was attributed to the upward gradient in
concentrations in the lower 1 km of the atmosphere measured in the early morning, and the
subsequent mixing across the planetary boundary layer (PBL) at midday. Similar results were
found by balloon soundings at the Vic Plain (N Barcelona) by Querol et al. (2017), and by
earlier studies of Millán et al. (1991, 1992, 1996a to c, 2000, 2002).
On the other hand, Gómez Moreno et al. (2011) and Brines et al. (2015) reported both
intensive summer and winter NPF episodes in the western border of Madrid City often with
the simultaneous occurrence of the highest $O_3$ episodes.
**2.2. Monitoring sites and instrumentation**
To characterise acute summer episodes of $O_3$ and UFP and to investigate their possible
relationships we devised an intensive field campaign in the MMA. Three measurement
supersites in and around Madrid, following a WNW direction, according the previously
described dynamics, were deployed in an area where the highest levels of $O_3$ are usually
recorded (Reche et al., 2017 submitted) inside the Madrid basin (Figure 1). Table 1 shows the
equipment available at the three following supersites:
•   Madrid-CSIC, located at the Spanish National Research Council headquarters. This site is
located in central Madrid on the sixth floor of the building of the Instituto de Ciencias
Agrarias.

•   CIEMAT, located at the Centro de Investigaciones Energéticas Medioambientales y
Tecnológicas headquarters, at 4 km in a WNW direction from the CSIC site in a suburban
area.

•   MJDH-ISCIII, located in the Instituto de Salud Carlos III in Majadahonda, at 15 km in a NW
direction from the CSIC site.

At MJDH-ISCIII, a PTR-ToF-MS has been deployed from 04 to 19/07/2017 and provides insights
into the $O_3$ Formation Potential (OFP) of the VOC mixture over the MMA area. The operation
procedure of the PTR-ToF-MS and OFP calculation are detailed in Table S1 and Figure S1.
Furthermore, from 11 to 14/07/2016, 28 profiles of pollutant and meteorological parameters
up to 1200 m a.g.l. were obtained using tethered balloons and a fast winch system (Figure S1,
Tables 2). The instrumentation attached to the balloons is summarised in Table 1. The profiles
were performed at the Majadahonda Rugby Course (MJDH-RC Figure 1). The balloons were
equipped with a Global Position System (GPS) and as set of the instruments (Figure S3),
including:



- A miniaturized CPC (Hy-CPC, Hanyang University) was used to measure number concentration of particles larger than 3 nm ($PN_3$) with a time resolution of 1 s and a flow rate of 0.125 L/min, using butanol as working fluid (Lee et al., 2014). Previous inter-comparison studies with conventional CPCs have yielded very good results (Minguillón et al., 2015). In this work, we will use the terms UFP and PN3 as equivalents but we measure concentrations between 3 and 1000 nm strictly while UFP is <100 nm. However, 80% of the total particle concentration falls in the range of UFPs.

- A PO3M $O_3$ monitor (2B Technologies) was used to determine $O_3$ concentrations. It was calibrated against an ultraviolet spectrometry reference analyser showing good agreement (n=34; $PO3MO_3=1.1058*RefO_3+4.41$, $R^2=0.93$). Concentrations (on 10 s basis) are reported in standard conditions (20 °C and 101.3 kPa) and corrected for the reference method.

In addition to the above instrumentation we obtained the following additional meteorological and air quality data:

- Meteorological data from the CIEMAT meteorological tower (four instrumented levels between surface and 54 m a.g.l.), as well as from several AEMET (Spanish Met Office) standard meteorological stations spread out across the basin: Madrid Airport (40.46°N, 3.56°W, 609 m a.s.l), Colmenar Viejo (40.69°N, 3.76°W, 994 m a.s.l), and El Retiro (in Madrid, 40.40°N, 3.67°W, 667 m a.s.l).

- Hourly data for air pollutants (NO, $NO_2$, $SO_2$, $O_3$, PM10 and PM2.5) supplied by the air quality networks of the city of Madrid, the Regional Governments of Madrid, Castilla la Mancha, Castilla y León, and the EMEP monitoring network, supplied by the National Air Quality Database of the Ministry of the Environment of Spain (MAPAMA).

- High resolution $O_3$-sounding data performed by AEMET at midday each Wednesday at Madrid Airport.

- High resolution meteorological sounding data obtained each day at 00:00 and 12:00 h local time by AEMET also at Madrid Airport. They were used to estimate the height of the PBL at 12:00 UTC by means of the simple parcel method (Pandolfi et al., 2014).

Hourly averaged wind components were calculated and used in polar plots with hourly $PM_1$, $PM_{2.5}$, $NO_2$, $O_3$, $O_x$ ($O_3+NO_2$), BC and UFP concentrations, by means of the OpenAir R package (Carslaw and Ropkins, 2012).

## 3. Results

### 3.1. Meteorological context

Figure 2 shows the time series of the recorded meteorology, measured at a surface station representative of the conditions in the MMA during the field campaign of July 2016 (El Retiro, in central Madrid). In order to put the field campaign into the context of the more general meteorological situation, the time series is extended backwards to the end of June and forward to the end of July 2016. Figure 2 also shows the corresponding time series of $O_3$, $NO_2$ and $O_x$ concentrations in the MMA, demonstrating the occurrence of well-marked peaks alternating with relatively low $O_3$ and $O_x$ concentrations periods. The intensive field campaign



(11-14/07/2016, marked with a green frame) coincides with a low $O_3$ interval preceding a
higher $O_3$ period in the last two days red and blue frames in Figure 2 show days in which high
resolution $O_3$ free soundings were performed (red and blue indicate intervals within high and
low $O_3$ respectively).
The AEMET $O_3$ soundings are represented in Figure 3 together with the maps of the 500 hPa
geopotential heights (gph in metres) and the MSLP (mean sea level pressure, in mb) contours
at 12:00 UTC obtained from the Climate Forecast System (CFS) reanalysis (Saha et al., 2014)
downloaded from http://www.wetterzentrale.de/. The low/high $O_3$ periods coincide with the
500 hPa gph passage of respectively upper level troughs/ridges over the area, associated with
cold/warm deep advection of air masses. Cold advections have usually an Atlantic origin.
The local meteorology during the field campaign was characterized by a progressive drop in T
(-4$^{\circ}$C in the maximal daily T) and an increase in the early morning RH (+20%), with insolation
remaining constant (maxima of 900-950 W/m$^2$) (Figure 4). During the nocturnal and early
morning conditions of the first half of the field campaign (11-12/07/2016), relatively weak
northerly winds prevailed at the main meteorological surface stations inside the basin,
including CIEMAT in Figure 4, and Retiro and Colmenar in Figure 5. This is probably related with
drainage (katabatic) conditions inside the Madrid basin, with a progressive turn to a more
synoptic westerly component in the central period of the day, consistent with a convective
coupling with the more intense upper level wind. This coupling is also accompanied by an
important increase of the wind speed at midday, up to 8 m/s (venting stage), that renewed air
masses in the whole basin. During the second half of the campaign, intense and persistent
north-easterly winds replaced the westerlies from the evening of 12/07/2016, after the
evolution of the upper level trough. In contrast to the previous period, during 13-14/07/2016
night-time and early morning conditions registered more intense NE winds (up to 10 m/s) than
at midday, after a decrease in intensity down to calm conditions (1 m/s) during the 12/07
morning facilitating both fumigation from upper levels and local $O_3$ photochemical production.
A weak wind veering to the south was also registered at the mentioned surface stations during
the 13/07 afternoon, which lasted only for 3 hours, and which is more characteristic of an $O_3$
enrichment episode, when the veering lasted longer (Plaza et al., 1997). A progressive
decrease of the PBL height (-600 m difference) is observed in the AEMET daily radio-soundings
that showed a gradual decrease of the midday PBL height, with 3400, 2200, 1900 and 1600 m
a.s.l. from 11 to 14/07/2016, Figure S3). This decrease is also observed in the 12 and
14/07/2017 UFP profiles (Figures 7-9, 11). As will be detailed later these meteorological
patterns allow $O_3$ and UFP to smoothly and progressively accumulate in the basin (Figure 4)
during the campaign.
In the vertical dimension, during both high and low $O_3$ periods analysed here, all the soundings
show at midday two well defined layers separated by a temperature inversion marking the
limit of the growing convection inside the PBL (Figure 3).
In high $O_3$ periods (6 and 27/07/2016) we found lower PBL heights (approximately 1300-1500
m a.s.l.), with weak winds from the E or NE (less than 4-5 m/s) or calm conditions. This is
consistent with the scheme proposed by Plaza et al., (1997), who also described a rapid
evolution of the PBL height up to 2500-3000 m a.s.l. at 15:00 UTC during their field campaigns
in the area under "summer anticyclonic conditions". They also described a morning radiative



surface inversion at around 1000 m a.s.l., which was usually "destroyed 1 hour after dawn",
and containing NE winds associated with nocturnal drainage flows at lower levels (following
the slope of the Madrid basin). In this context, residual layers containing pollutants processed
during the previous day can develop above the stably stratified surface layer during night-time
conditions. These pollutants can be transported towards the S by weak north-easterly winds,
or either remain stagnant under calm conditions and lead to fumigation and mixing with fresh
pollutants emitted at the surface after the destabilization of the surface layer as we evidenced
in our profiles. These residual layers are topped by the subsidence anticyclonic inversion
(1000-1500 m a.s.l.) according to Plaza et al. (1997).
Conversely, the soundings corresponding to low $O_3$ periods have in common more elevated
PBL heights (2000-2500 m a.s.l) with more intense winds (above 6-7 m/s) that can blow from
different sectors: from the NE, on the 13/07/2016 (with intense N-Westerlies blowing in the
free troposphere) or from the S-SW as observed on the 29/06/2016 and 20/07/2016. The $O_3$
sounding on 13/07/2016, a unique day within the field campaign, presents the final stage of a
low $O_3$ period with winds in the free troposphere with a clear NW component while channelled
north-easterly winds dominate below 2000 m a.s.l. The decrease of surface temperature
observed in Figure 2 during the field campaign, is also consistent with the cold advection
associated with the troughing in the 500 hPa heights.
**3.2. Surface $O_3$, $O_X$ and UFP during the field campaign**
Figure 4 shows the time series of meteorological parameters (CIEMAT tower), $NO_2$, $NO$, $O_3$, BC
and UFP concentrations at Madrid-CSIC, Madrid-CIEMAT and MJDH-ISCIII, as well as at MJDH-
RC, for the period 11-15/07/2016. As previously stated, the field campaign was characterised
by atmospheric venting conditions with the two latter days being in the transitional period to a
more stable anticyclonic episode of increasing $O_3$. The lowering of the wind speed during
diurnal periods and other meteorological features mentioned above favoured the gradual
accumulation of pollutants as indicated by the progressive increase of the $O_3$ maxima at MJDH-
ISCIII where the $O_3$ maximum was reached at 15:00 UTC on 13/07/2016 and at 17:00 UTC on
14/07/2016 (Figure 4). The typical accumulation $O_3$ cycle for the zone was found only on 13
and 14/07/2016 with a maximum at 14:00 UTC on 13/07/2016 and at 16:00 UTC on
14/07/2016. The two previous days presented a more irregular daily pattern, indicating
unstable and atypical situations for July (perturbed conditions with prevalence of the synoptic
winds). Furthermore, these meteorological conditions and the high insolation induced the
concatenation of nucleation episodes in the basin (with low BC and very high UFP levels at the
central hours of the day), such as the one on 13/07/2016 (Figure S5).
From 11 to 12/07/2016 the highest concentrations of $O_3$ were recorded for W-SW and W
winds, and peak UFP ($PN_3$) concentrations were observed with W, SW, WNW and NE winds;
however on 13-14/07/2016 both $O_3$ and UFP concentrations maximized during calm and NE
winds (Figure 6). $PM_{2.5}$ levels were independent of the UFP and $O_3$ variation, increasing in calm
situations in the first two days, and with less variation but with somewhat higher
concentrations with NE winds in the last two days (Figure 6).
In Figure S5 the evidence for the occurrence of a NPF episode on 13/07/2016 is shown.
Morning-midday UFP bursts were caused by nucleation and growth episodes. As previously





stated, in a twin article we will focus on the phenomenology and the vertical occurrence of
these nucleation-growth events.

### 3.3. Vertical $O_3$ and UFP profiles during the field campaign

Considering the $O_3$ profiles in Figure 3, high $O_3$ concentrations (greater than 70 ppb) can be
observed above the PBL, between 3000 and 5000 m a.s.l., which may be related to larger scale
transport of pollutants previously uplifted to the mid-troposphere. However, at lower levels
(inside the PBL) the higher concentrations correspond to the accumulation days (06 and
27/07/2016). As will be demonstrated in this section, $O_3$ concentrations within the PBL
increase throughout the day under all atmospheric conditions due to fumigation from the
residual layer, and new $O_3$ formation from fresh precursors emitted at night-time and through
the day. However, larger increases of $O_3$ concentrations were registered on poorly ventilated
days.
As shown in Figure S2 and Table 2, the vertical profiles for 14/07/2016 were the most
complete of the campaign (wind speed was relatively low and this allowed extended
measurements along the day), and for that reason we begin with the description on this day.
Figure 7 shows that there is a rapid growth of the PBL between 08:05 and 11:01 h UTC, as
deduced from the vertical profile of UFP ($PN_{3-300}$) concentrations. At the beginning of the
measurements the upper limit of the PBL was above 1030 m a.s.l. and in 2 h 40 min it was
lifted 400 m (around 2.5 m/min). In this initial period, the vertical profile of $O_3$ was
characterized by a succession of strata of different concentrations, but with a clear tendency
to increase towards higher altitudes (around 20 ppb of difference between surface level and
1950 m a.s.l. was observed). The discontinuity of the PBL ceiling reflected in the UFP, T and RH
profiles did not seem to affect at all the $O_3$ profile. In other words, we did not notice
accumulation of $O_3$ layers in the top of the PBL, but a general trend to increase towards the
highest altitudes reached with the tethered balloons.
Through the course of the day the profile of concentrations of UFP and $O_3$ became
homogenous in the lowest 1200 m a.g.l. (this being the maximum height reached), and a
growth of $O_3$ concentrations at all altitudes was observed until 16:11 h UTC. This
homogenisation and growth of $O_3$ concentrations in the PBL, caused by intense mixing by
convection, resulted in a an uneven growth through the day with an increase of 43 ppb at
surface and only 10 ppb at 1900 m a.s.l. (Figure 8).
Figure 9 shows the results from measurements taken at a fixed height (1400-1200 m a.s.l.)
made to capture the effect of the growth of the PBL on $O_3$ and UFP levels. We started at
around 700 m a.g.l. at 09:32 UTC with 60 ppb of $O_3$ and around 6000 #/cm³. At 10:25 UTC the
top of the PBL reached the balloon as deduced from the sharp increase in UFP concentrations
(up to 20000 #/cm³). Meanwhile, $O_3$ concentrations experimented only a slight decrease
suggesting that $O_3$ fluxes are top down and not bottom up as recorded for UFP. From 16:11 h
UTC onwards, a reduction of $O_3$ levels at lower heights was observed (-50 ppb at surface levels
from 15:55 to 17:45 h UTC while at 1900 m a.s.l. levels remained stable, Figure 8).
The first balloon flight on 13/07/2016 was performed at 10:45 UTC because earlier the wind
speed was too high (Figure 10). At that time the top of the PBL had developed beyond the
maximum height reached with the tethered balloons, so in the profile above 1100 m a.g.l. a





very homogeneous concentration was detected. At this time on 14/07/2016 the upper bound
of the PBL was perfectly identifiable in the UFP vertical profile over 700 m a.g.l., thus the
growth of the PBL was faster on 13/07/2016 than on 14/07/2016. Similarly to 14/07/2016, the
13/07/2016 $O_3$ profiles were characterised by a progressive increase of concentrations with
height (more accentuated in different strata). The profiles started with concentrations close to
40 ppb $O_3$ at the surface, and reached 83 ppb at the upper heights. As occurred on
14/07/2016, through the course of the day surface concentrations increased differentially with
respect to the upper layers, to almost homogenize concentrations in the whole profile
(between 68 and 80 ppb at all heights at 15:00 UTC).
In Figure 11 it can be observed that similar results to those described for UFP profiles on the
14/07/2016 were found on 12/07/2016 (upwards growth of the top of the PBL from the early
morning):
• Around 700 m a.g.l. at 07:30 UTC (5000 #/cm$^3$ surface concentrations, 2000 #/cm$^3$ at the
top of the PBL, and 900 #/cm$^3$ in the free troposphere).

• Around 900 m a.g.l. at 09:00 UTC (9000, 5000 and 2000 #/cm$^3$ for the above three levels).
• Above 1200 m a.g.l. (this being the maximum measurement height) at 10:00 UTC (10000
#/cm$^3$ surface concentrations and 7000 #/cm$^3$ at 1200 m a.g.l.) and 12:55 y 13:42 h (10000
#/ cm$^3$ surface concentrations and 20000 #/cm$^3$ at the maximum height of 900 m a.g.l.).

In the early morning of 12/07/2016 $O_3$ strata at different heights within the PBL were
detected, with concentrations reaching 30 to 55 ppb and higher levels (55 to 65 ppb) at the
highest altitude reached. During the 10:00 UTC flight $O_3$ levels reached 75 ppb at the top level
decreasing gradually down to 40 ppb at surface levels. At 12:00 UTC concentrations at the top
of the profile reached 87 ppb, 70-75 ppb in the 100-700 m a.g.l. transect and 60 ppb in the
lowest 100 m a.g.l., where NO titration and $O_3$ deposition was more efficient. Thus, the
12/07/2016 profiles again showed a vertical trend characterised by i) higher $O_3$ concentrations
at the highest sounding altitude in the early morning, ii) increase in $O_3$ concentrations as the
morning progressed (more pronounced at low altitudes), and iii) homogenous $O_3$
concentration along the entire vertical profile, except in the surface layers, where the
deposition and titration markedly decreased $O_3$ levels reached at midday. These vertical
trends, with concentrations exceeding 75 ppb $O_3$ above 100-250 m a.g.l., and a marked
decrease down to 60 ppb at surface levels was also evident during the short profiles obtained
on 11/07/2016 at 18:28-18:41 UTC (Figure 11).

## 4. Discussion

According to the $O_3$-soundings and radio-soundings analysed above, as well as previous
evidence described by Plaza et al. (1997) and the surface air quality measurements presented
in this study, surface $O_3$ formation from precursor emissions within the MMA seems to
develop in the core of regional processes, modulated by large scale meteorological conditions,
distinguishing two types of episodes:
• ACCUMULATION, occurring in stable stagnant conditions and regional accumulation of
pollutants (in the sense of Millan et al., 1997, 2000; Gangoiti et al., 2001, Millán, 2014), with
high $O_3$ reserve strata accumulated during the previous day in the residual layer and



associated with fumigation around midday in the following day. The $O_3$ concentrations are
high along the whole atmospheric column, but enriched in the lower section by additional
local formation of $O_3$ within the PBL and transport-recirculation of the urban plume of
Madrid around the area. This transport is characterised by a net transport to the NW-N
during daytime, after vertical mixing, and to the S and SW during night-time, inside the
residual layer and decoupled from a more stable nocturnal surface layer. Typically
pollutants accumulate during periods of 2-6 days resulting in a well-marked peak and valley
concentration periods that affect background, peri-urban and in-city stations. This is the
case for the $O_3$-soundings of 29/06/2016 (not shown), and particularly 27/07/2016 (Figure
12), or the measurements with captive and free balloons by Plaza et al. (1997) in 1993 and
1994, with very high concentrations of $O_3$ in the lower atmospheric layers, usually forming a
bump in the vertical profile of $O_3$, below a height of 2000 m a.s.l., easily reachable after
daytime convection (Figure 12). As illustrated for 06/07/2017, OFP (Table S1 and Figure S1)
may be largely dominated by the carbonyls (mostly formaldehyde and acetaldehyde),
followed by aromatic compounds (benzene, boluene, C8 aromatics, C9 aromatics and C10
aromatics) when considering the VOC pool during the morning traffic peaks. The influence
of aromatic VOCs on OFP rapidly decreases while the influence of biogenic VOCs (primary
and secondary) increase throughthe day resulting in a similar potential influence of
biogenic and aromatic VOCs on $O_3$ formation during accumulation periods, but with an OFP
still dominated by carbonyls (see supplementary information for additional supporting
material).
• VENTING, occurring in advective atmospheric conditions (in the sense of Millan et al., 1997,
2000; Gangoiti et al., 2001, Millán, 2014) with $O_3$-soundings characterized by (probably
external) contributions from high altitude $O_3$ strata, and their fumigation on the surface
(episodes 11-14/07/2016). There is no accumulation of pollutants above the stable
nocturnal boundary layer, if any, because more intense and steady winds are charged to
sweep out the local production during the preceding day. OFP contributions of carbonyls
(dominating OFP), aromatic and biogenic VOCs did not significantly vary for 13 and
14/07/2017 from what it is described above for 06/07/2017.
As detailed in sections 3.1 and 3.2, with weakening of general atmospheric circulation by the
end of the campaign period, $O_3$ and UFP smoothly and progressively accumulated in the basin
(Figure 7). An observed decrease of the PBL depth (up to -1800 m at midday according AEMET
radio-soundings during the campaign Figure S4), probably also contributed to the progressive
increase of pollutant concentrations through the campaign.
With respect to the vertical variability, the general pattern for UFP ($N_3$) clearly showed a rapid
and marked growth of the PBL in the first hours of daylight (Figure 13). In these early stages of
the day, $O_3$ profiles were characterized by a succession of strata of different concentrations,
but with a clear increasing trend towards the higher levels (Figure 13). The discontinuity of the
PBL ceiling, reflected in the UFP, temperature and humidity profiles, was not identified as such
in the $O_3$ profiles (Figures 7, 9 and 11). As the day progresses the UFP and $O_3$ concentration
profiles are homogenized and a progressive diurnal growth of $O_3$ concentrations occurs until
16:00 or 17: 00 UTC (Figure 13) which is observed most clearly at the surface. This vertical
variability points to different aspects such as: (i) the relevance of fumigation from high altitude
$O_3$-rich strata; ii) surface titration and deposition of $O_3$; (iii) surface photochemical generation





of $O_3$ from precursors (with higher concentrations close to the surface); and (iv) horizontal $O_3$
and precursor surface transport from the urban plume of Madrid towards MJDH-RC. The upper
$O_3$-rich strata might have an external (to the Madrid basin) origin, or might have been injected
regionally at high altitudes on the previous day(s) by the complex re-circulations of air masses
already reported by Millán et al. (1997, 2000, 2002); EC (2002 and 2004); Gangoiti (2001),
Mantilla et al. (1997), Castells et al. (2008a and b) and Millán (2014) for the W Mediterranean,
by McKendry et al. (2000) for other parts of the world; and by Plaza et al (1997), and Diéguez
et al. (2007 and 2014) for the Madrid area.
According to the last referenced authors, due to the orientation of the Sierra de Guadarrama
(Figure 1) the heating of its S slopes throughout the day forces the wind direction to veer,
describing an arc that sweeps the zones to the N of Madrid clockwise, from the W to the NE.
Dieguéz et al. (2014) showed that the $O_3$ maxima are recorded at an intermediate point on this
route (El Pardo, Colmenar V., see location in Figures 14 and S6) determined by the wind speed,
the initial composition of the urban plume, and the result of photochemical processes on its
route from the metropolitan area to tens of kilometres away. In addition, our results and those
of Plaza et al. (1997) show that $O_3$ fumigation from high atmospheric layers decisively
contributes to the increases in the surface levels, since surface concentrations during our
measurements never exceeded those recorded at the highest altitude reached, and at midday
homogeneous $O_3$ levels are measured across the lower 1.2 km of the PBL. During the whole
month of July 2016 the described veering of the urban plume, towards W (MJDH-San Martin
de V., see location in Figures 14 and S6) in the early hours, and towards NW, N-NE, and, in
some cases E and SE, followed by the decoupling and onset of the nocturnal flow towards SW,
seems to be causally associated with the $O_3$ information threshold exceedances, since the
maps of exceedances recorded by the official air quality network follow this spatial and
temporal evolution (Figure S6). These plume impacts occur in periods when the $O_3$
concentration is already high because of accumulation in the air mass from one day to the next
which is not completely renewed due to general circulation conditions. The relevance of the
latter has been recently demonstrated by Otero et al. (2016) who reported the maximum
temperature as the parameter more directly related with high $O_3$ concentrations in central
Europe, whereas in the Mediterranean regions it was a high $O_3$ concentration recorded with a
lag of -24 h.
The differential afternoon-evening decrease of $O_3$ surface concentrations compared with those
found at the top of the flights can be attributed to (i) the lower intensity or weakening of the
fumigation processes; (ii) a greater $O_3$ titration and deposition in the lower PBL; and (iii) the
lower photochemical $O_3$ production after the midday insolation maxima. Thus, this process
again demonstrates the relevance of high altitude layers and their fumigation to the surface, in
the hours of maximum convection.
Regarding the concentrations of UFP, they were very homogeneous throughout the PBL during
the vertical profiles, especially in the hours of maximum convection, showing a marked
increase from 11 to 14/07/2016 in the whole depth for all profiles (Figure 13). Thus, on the
12/07/2016, the upper limit of the PBL (marked by a sharp reduction of UFP levels) reached
900-1200 m a.g.l. respectively in the flights conducted at 08:05 and 10:12 UTC (Figure 13). In
turn, on the 14/07/2016, the top of the PBL at midday exceeded 1200 m a.g.l. only in the
afternoon, being constrained to 300 to 700 m a.g.l. from 08:05 and 10:45 h UTC (also shown in





the progressive loss of -1800 m in the midday PBL height from 11 to 14/07/2016, revealed by
AEMET radio-soundings).
The enhanced convection on the 12/07/2016 probably favoured the dilution of UFP
concentrations and reinforced the fumigation of $O_3$ from upper levels. Conversely, the lower
development of the PBL on 14/07/2016 hindered the fumigation of upper $O_3$ layers, resulting
in an opposite temporal trend for $O_3$ and UFP along the profile. Thus, a weaker development of
the PBL might result in the increase of UFP concentrations, even if UFP emission/formation
rates did not vary significantly. However, we cannot discard the possibility that this UFP
increase on the last day was the result of a higher intensity and duration of the nucleation
episodes.
Consideration of the evolution of surface $O_3$ concentrations (as shown in Figure 14, on the 11
and 12/07/2016) depicts a double wave: the first peak around midday (11:00-14:00 UTC on the
first day, and 12:00-13:00 on the second) and the second one in the afternoon-evening (19-
22:00 and 16:00-20:00 UTC, respectively), showing relative peaks (not always, sometimes just
a plateau). We interpret that the morning increase of $O_3$ concentrations is dominated by both
local production, still dominated by anthropogenic VOCs (Figure S1), and fumigation of upper
levels, with an early maximum when layers above are rich in $O_3$, which progressively decrease
with dilution with surface concentrations. The secondary evening concentration peak
corresponds to the advection of a locally enriched $O_3$ air mass (titration always causes $O_3$
depletion towards nocturnal values). When both processes (morning fumigation and evening
advection) are not so strong, $O_3$ local production results in a more typical diurnal time
evolution with a single maximum at 15:00-16:00 UTC on 13-14/07/2016 (Figure 14).
The relative importance of the local contribution of the MMA to the $O_x$ concentrations
registered in the monitoring stations has also been elucidated by comparing the observations
at upwind and downwind locations relative to the city. At this respect, Atazar and Alcobendas
(Figure 14) are located downwind for 11 and 12/07/2016 and MJDH and Fuenlabrada are
upwind while the opposite occurs for 13 and 14/07/2016. As the urban air mass is transported
towards the E and NE during the first two days, a local $O_x$ contribution is superimposed to the
background at Atazar and Alcobendas where recorded $O_x$ was the highest in the basin (Figure
14). The contrary holds during the next two days, when these sites show lower concentrations
than the rest. MJDH and Fuenlabrada show a reversed behaviour, with lower concentrations
during the first two days and higher for the last days.

**5. Conclusions**
The phenomenology of $O_3$ episodes in the Madrid Metropolitan Area (MMA, Central Iberia)
has been characterised. We found that $O_3$ episodes linked with precursors emitted in the
Madrid conurbation are modulated by the complex regional atmospheric dynamics.
Vertical profiles (up to 1200 m a.g.l.), obtained using tethered balloons and miniaturised
instrumentation in Majadahonda (MJDH), a sub-urban site located on the southwestern flank
of the Madrid Metropolitan Area (MMA) during 11-14/07/2016, showed how critical processes
developed with altitude. Simultaneously, measurements of a number of air quality and



meteorological parameters were carried out at 3 supersites within the MMA, where spatial
differences highlight the influence of atmospheric dynamics at different scales.
The results presented here confirm prior findings regarding the concatenation of relative low
(venting) and high (accumulation) $O_3$ episodes in summer. In the MAB, during both types of
episodes, fumigation of high altitude $O_3$-rich layers contributes with a relevant proportion to
surface $O_3$ concentrations. Moreover, we propose here a conceptual model shown in Figure
15. Particularly, accumulation episodes are activated by a relatively thinner PBL (< 1500 m
a.g.l. at midday), low synoptic winds, and the development of anabatic winds along the slope
of the Sierra de Guadarrama (W and NW of MAB, with >2400 m a.g.l. peaks). This PBL height,
lower than the mountain range, and the development of the mountain breezes cause the
vertical recirculation of air masses and the enrichment of $O_3$ in the lower troposphere as well
as the formation of reserve strata that fumigate to the surface as the diurnal convective
circulation develops. These atmospheric dynamics account for the occurrence of the high Ox
($O_3$+$NO_2$) surface concentrations. During venting episodes with a more intense synoptic wind
and the top of the PBL usually reaching >2000 m a.g.l, vertical $O_3$ profiles were characterised
by an upward increase of concentrations, whereas lower altitude $O_3$ maxima were observed in
the accumulation periods. Interestingly, vertical profiles demonstrated that during the study
period $O_3$ fumigation (top-down flow) from upper layers prevailed as a contribution to surface
$O_3$ concentrations, whereas the increase of UFP takes place bottom-up, progressing with the
development of the PBL and the occurrence of nucleation and growth episodes occurring
within the PBL. Thus, when crossing the boundary of the PBL from the free troposphere
increases of UFP concentrations by an order of magnitude and a slight decrease of $O_3$ levels
were registered. This $O_3$ and UFP vertical distribution through the day is consistent with the
existence of an efficient venting mechanism, which is able to sweep out the local production of
the day. Thus, there is no accumulation of pollutants above the observed stable nocturnal
boundary layer from one day to the next, and new UFP production is added from below the
following day. The presence of $O_3$ enriched layers well above the stable nocturnal boundary
layer suggests a remote origin of this pollutant in photochemical processes developed at least
the day before away from the Madrid basin.
The results obtained in this intensive field campaign can be summarized in the following
conclusions and recommendations concerning $O_3$ abatement policies:
•   The phenomenology of $O_3$ episodes in S Europe is extremely complex, mainly due to
the close relation between photochemistry processes and mesoscale atmospheric
dynamics, requiring, consequently, abatement policies very different to the ones
useful for Central Europe, as intensive research has demonstrated in the last decades.

•   During the highest $O_3$ (accumulation) episodes, apart from the fumigation contribution
(X in Figure 15) to surface $O_3$ concentrations there is an added fraction of $O_3$ produced
locally or transported horizontally (Y in Figure 15). If sensitivity analyses demonstrate
that abatement of specific precursors would have an effect reducing $O_3$ peaks, the
reduction strategies (geographic extension, timing…) to decrease Y and X components
are very different, and, in most cases, the X component will dominate in the relative
contributions. Thus, probably, structural measures over wider regions would be more
effective than episodic tactics that might have a larger effect on the Y component. In



terms of precursors, the OFP analysis carried out at ISCIII site shows that even if
anthropogenic emissions may still dominate the $O_3$ formation through the potential
impact of alkenes and alkanes (not measured here) and the high contribution of
carbonyls (formaldehyde and acetaldehyde), biogenic emissions must be considered.
Biogenic VOC (primary and secondary) and aromatic compounds (C6 to C10)
contribute to the same extent to the OFP.

•    The meteorological scenarios causing the summer accumulation episodes in the MAB
(high temperatures, low synoptic winds and relatively thinner PBL) should be forecast,
to drive an effective alert system on the possible occurrence of pollution episodes.

•    It is necessary to achieve a more detailed characterisation of $O_3$ precursors (VOC and
biogenic VOCs, BVOCs) in the MAB, especially in the source areas, to effectively predict
the photochemical evolution of the plumes, and the main impact areas where $O_3$ from
high altitude layers formed the day(s) before from other precursors fumigates to the
surface levels enriched in $O_3$ and other precursors.

•    Sensitivity analyses using modelling techniques will permit simulation of the real
situation concerning the $O_3$ abatement potential but only if the following is achieved in
advance: i) reproduce the recirculation cells and other local/regional complex
meteorological patterns such as the fumigation processes and the plume transport; ii)
include a geographically resolved and accurate emission inventory of $O_3$ precursors in
the source areas; and iii) reproduce the origin of the high altitude $O_3$ strata from
external origins.

The conceptual model described here for $O_3$ episodes confirming the relevance of the vertical
re-circulations that Millan et al (1997, 2000), Gangoiti et al. (2001) and Millán (2014)
highlighted, and controlled in this case by specific synoptic conditions the PBL depth, may be
also applicable to most S Europe. Thus, Otero et al. (2016) demonstrated that in central Europe
the highest temperature is the most statistically related parameter for $O_3$ episodes, while in S
Europe it is the $O_3$ levels recorded the day before (reflecting re-circulation).

## Acknowledgements

The present work was supported by the Spanish Ministry of Agriculture, Fishing, Food and
Environment, Madrid City Council and Madrid Regional Government, and by the Ministry of
Economy, Industry and Competitiveness and FEDER funds under the project HOUSE (CGL2016-
78594-R), and by the Generalitat de Catalunya (AGAUR 2015 SGR33). Part of this research was
supported by the Korea Ministry of Environment through "The Eco-Innovation project". The
support of the CUD of Zaragoza (project CUD 2016-05), UPV/EHU (UFI 11/47, GIU 16/03), the
project PROACLIM CGL2014-52877-R, the City Council of Majadahonda for logistic support and
AEMET for providing surface meteorological data, and data from radiosoundings and ozone
free-soundings is also acknowledged. We thank Alava Ingenieros, TSI, Solma Environmental
Solutions, and Airmodus for their support, and  María Díez for her computer support in the
treatment of radiosonde data.



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



**FIGURES AND TABLES**

**Figure Captions**
Figure 1. Location of the study area, profiles showing the major orographic patterns and
location of three supersites (CSIC, CIEMAT, ISCIII) and the site were vertical profile
measurements were carried out (MJDH).
Figure 2. Top: Hourly meteorological parameters recorded at El Retiro air quality monitoring
station in central Madrid (from 28/06/2016 to 01/08/2016). Middle: Hourly concentrations of
$O_3$ and $O_X$ ($O_3$+$NO_2$) recorded at a selection of air quality monitoring station representing the
Greater Madrid area, together with those from the remote background station of
Campisábalos. Bottom: Hourly $NO_2$ concentrations recorded at the same sites for the same
period. Periods with available AEMET free-soundings of $O_3$ are bracketed with red
(accumulation) or blue (venting) squares. The vertical $O_3$ and UFP profiling campaign is marked
with a green square.
Figure 3: Left: Climate Forecast System Reanalysis (CFSR) for the 500 hPa geopotential heights
(gpdams) and mean sea level pressure (MSLP) contours (hPa) at 12:00 UTC in July 2016
(Wetterzentrale, http://www.wetterzentrale.de/), simultaneous with Right: AEMET $O_3$-free
soundings at Madrid airport.
Figure 4. Variation of meteorological parameters (temperature, relative humidity, solar
radiation and wind speed and direction), and levels of $NO_2$, NO, $O_3$, PM2.5, PM1, BC and UFP
(with lower detection limits of 1, 3 and 7 nm, $PN_1$, $PN_3$ and $PN_7$) measured at Madrid-CSIC,
Madrid-CIEMAT and ISCIII, as well as in MJDH-RC from 11 to 14/07/2016.
Figure 5. Wind roses for Madrid-CIEMAT and AEMET (El Retiro and Colmenar Viejo stations)
and location of the vertical profiling site (MJDH-RC).
Figure 6. Polar plots of the concentrations of hourly $O_3$ (upper), UFP ($PN_3$, medium) and PM2.5
(lower) concentrations measured at Madrid-CSIC, Madrid-CIEMAT and MJDH-ISCIII from 11 to
14/07/2016. Wind data used in all cases is the one from the CIEMAT meteorological tower.
Figure 7. Vertical profiles of levels of $O_3$, UFP ($PN_3$), temperature and relative humidity
obtained on 14/07/2016 (8:05 to 17:45 UTC). A: Ascending; D: Descending.
Figure 8. Vertical profiles of levels of $O_3$, UFP ($PN_3$), temperature and relative humidity
obtained on 14/07/2016 (8:05 to 17:45 UTC), showing a top-down growth of differential $O_3$
concentrations from 08:05 with respect those from 15:55 UTC, as well as a bottom up
decrease of this differential concentration between 15:55 and 17:45 UTC. A: Ascending; D:
Descending.
Figure 9. UFP ($PN_3$) concentrations for different vertical profiles obtained on 14/07/2016, as
well as $O_3$ and UFP during two periods focusing to evaluate changes produced in a fixed height
when reached by the growth of the PBL.
Figure 10. Vertical profiles of levels of $O_3$, UFP ($PN_3$), temperature and relative humidity
obtained on 13/07/2016 between 10:45 and 15:06 UTC. A: Ascending; D: Descending.



Figure 11. Vertical profiles of levels of $O_3$, UFP ($PN_3$), temperature and relative humidity
obtained on 12 and 11/07/2016. A: Ascending; D: Descending.
Figure 12. Top: Vertical profiles of $O_3$ levels, and temperature obtained on 12/07/1994 (with
free sounding) and 15/07/1993 (with tethered balloons). Data obtained from Plaza et al
(1997). Bottom: Vertical profiles of $O_3$ levels of the free soundings by AEMET at Madrid airport
(26.6 km east of MJDH-RC) in 06-07/2017.
Figure 13. 11-14/07/2017 profiles of $O_3$ and UFP ($PN_3$) grouped by hourly stretches from
morning to afternoon.
Figure 14. Time evolution of hourly $O_X$ ($O_3+NO_2$) and $O_3$ concentrations from 11 to 14/07/2016
at selected air quality monitoring sites of the Madrid Basin and an external reference site
(Campisábalos), as well as the locations of these monitoring sites.
Figure 15. Conceptual model of the venting and accumulation $O_3$ episodes in the Madrid Air
Basin, their associated vertical $O_3$ profiles and the X (fumigation from upper layers) and Y
(local/regional) contributions to surface $O_3$ concentrations in the accumulation episodes.





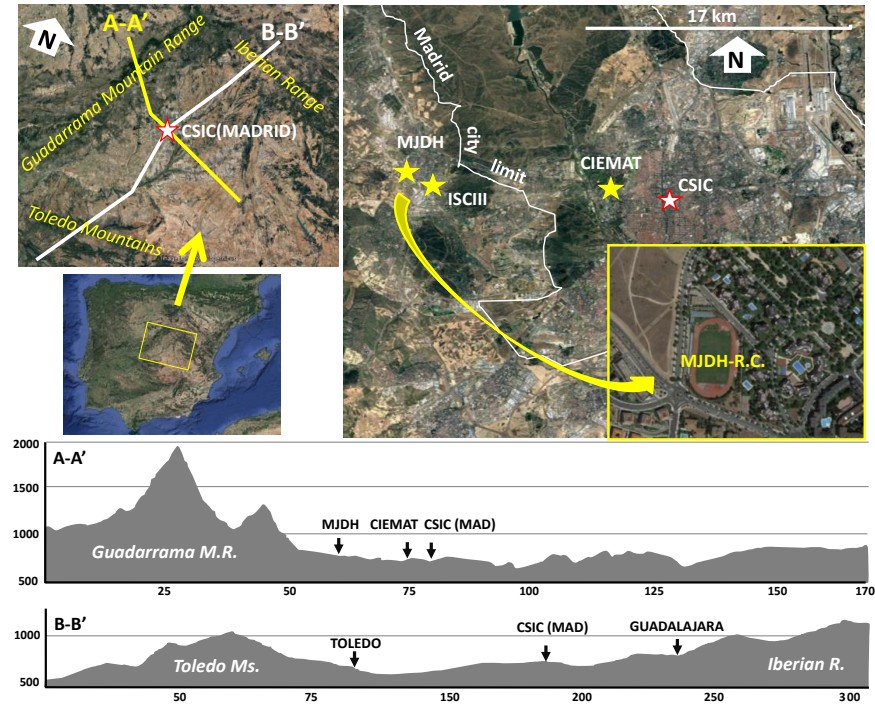

**Figure 1**



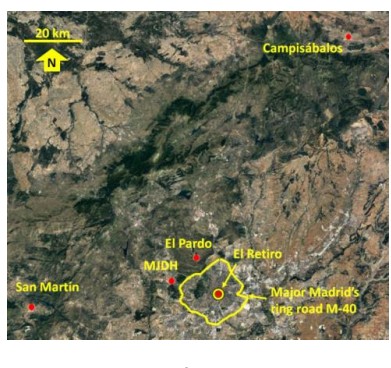

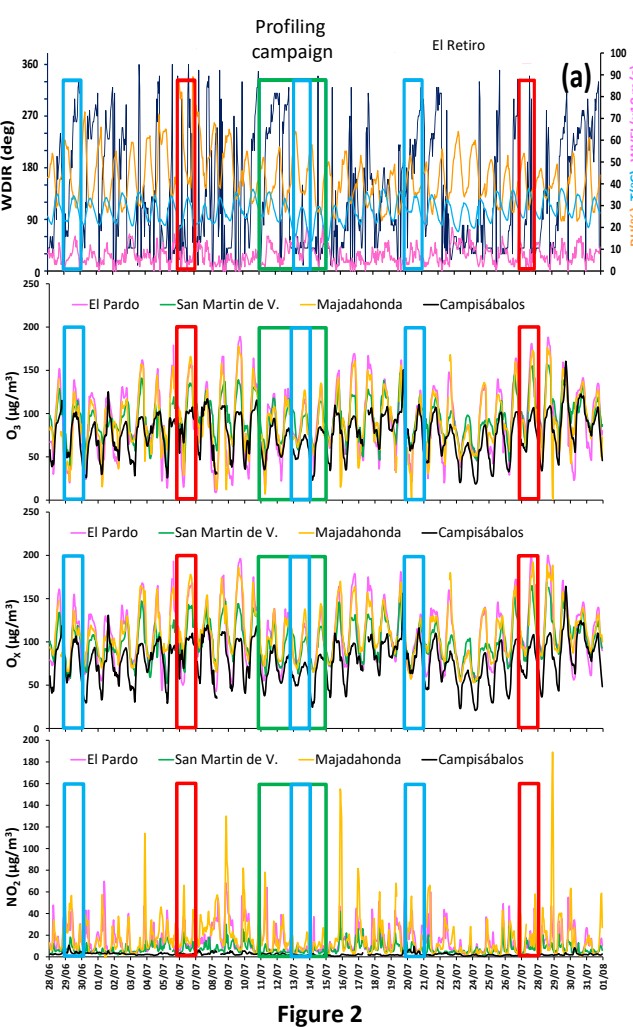


**Figure 2**





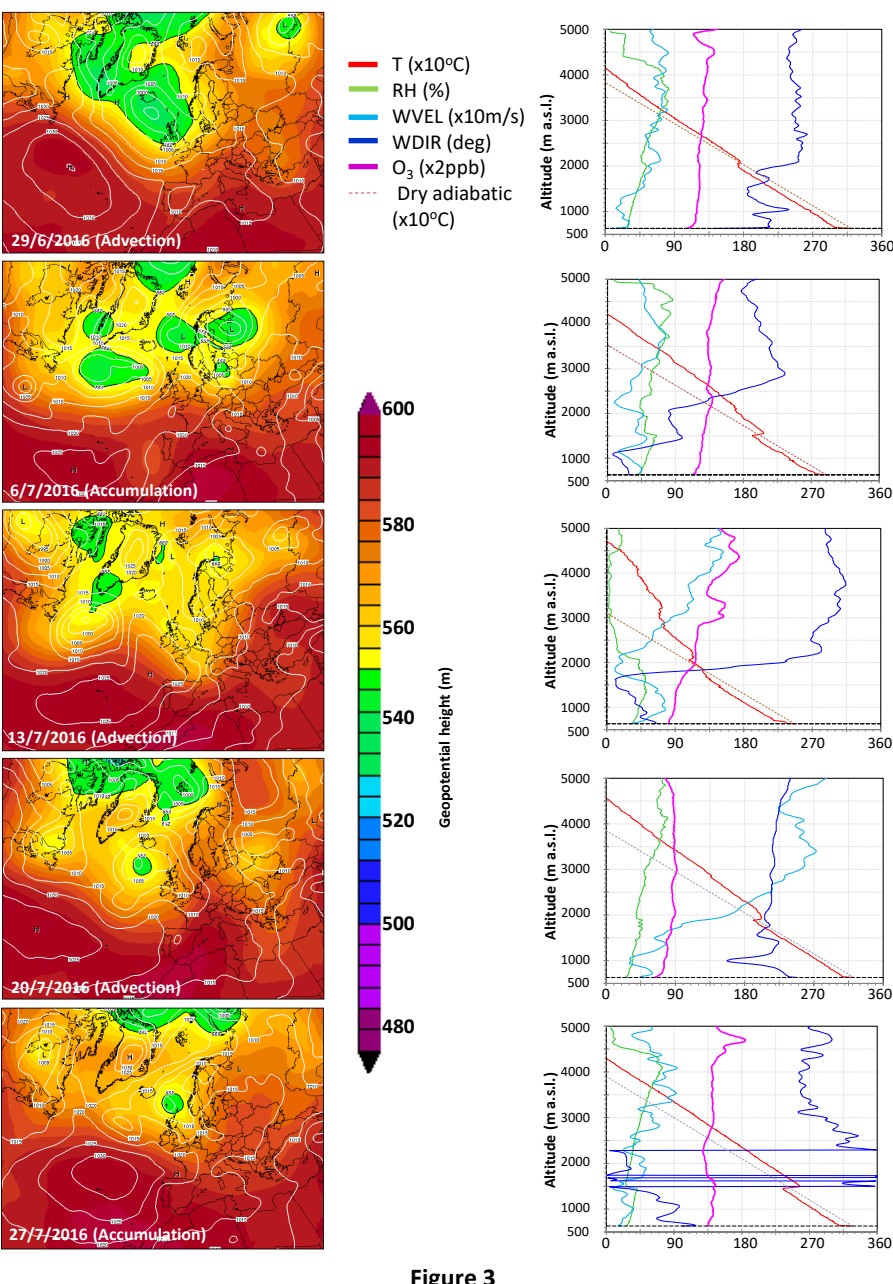

**Figure 3**




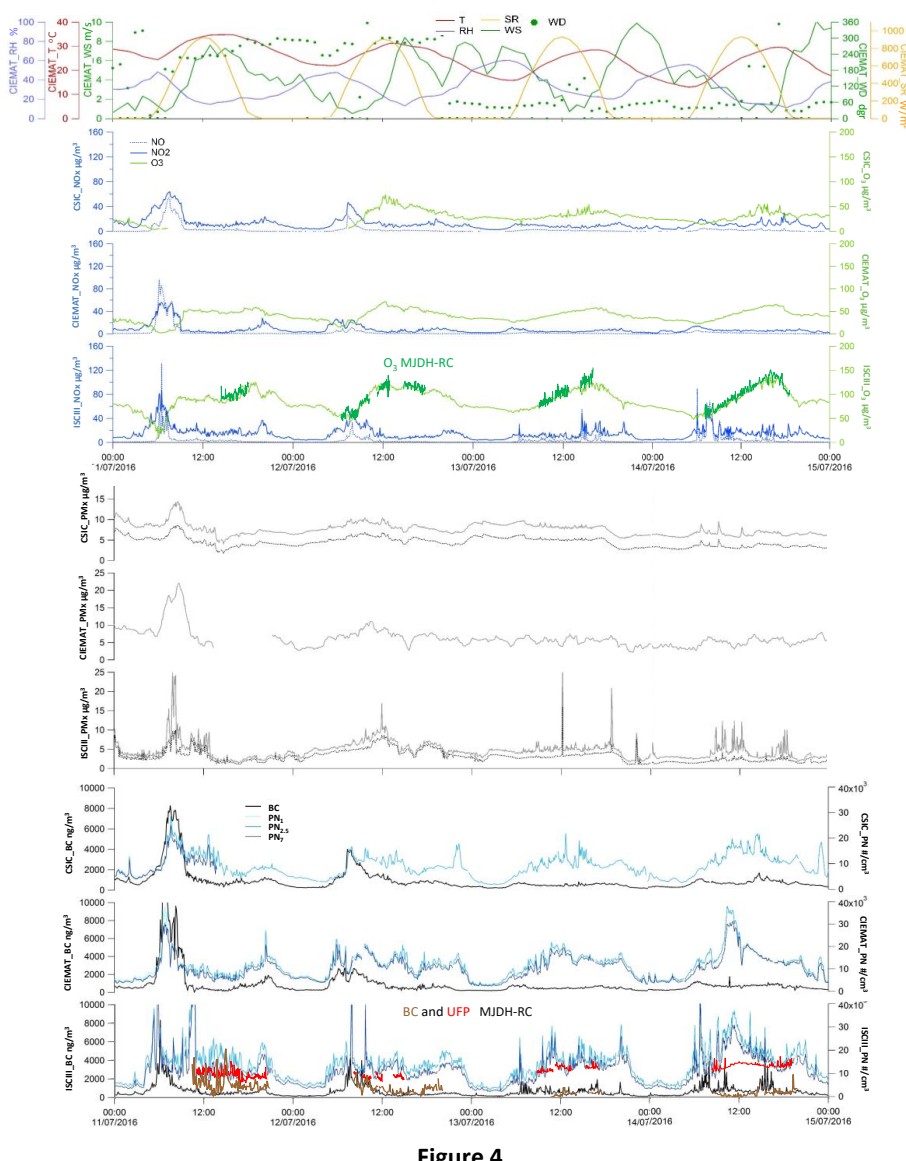

**Figure 4**





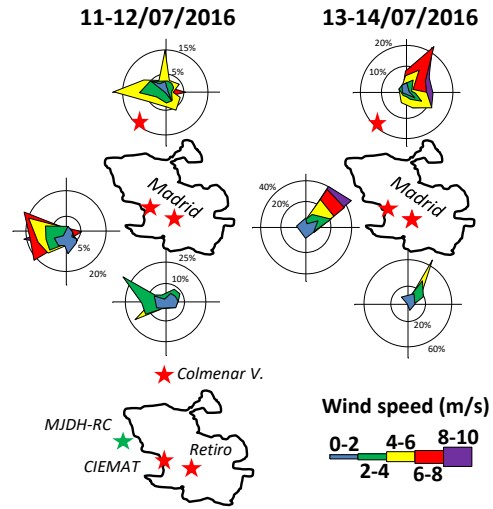

**Figure 5.**





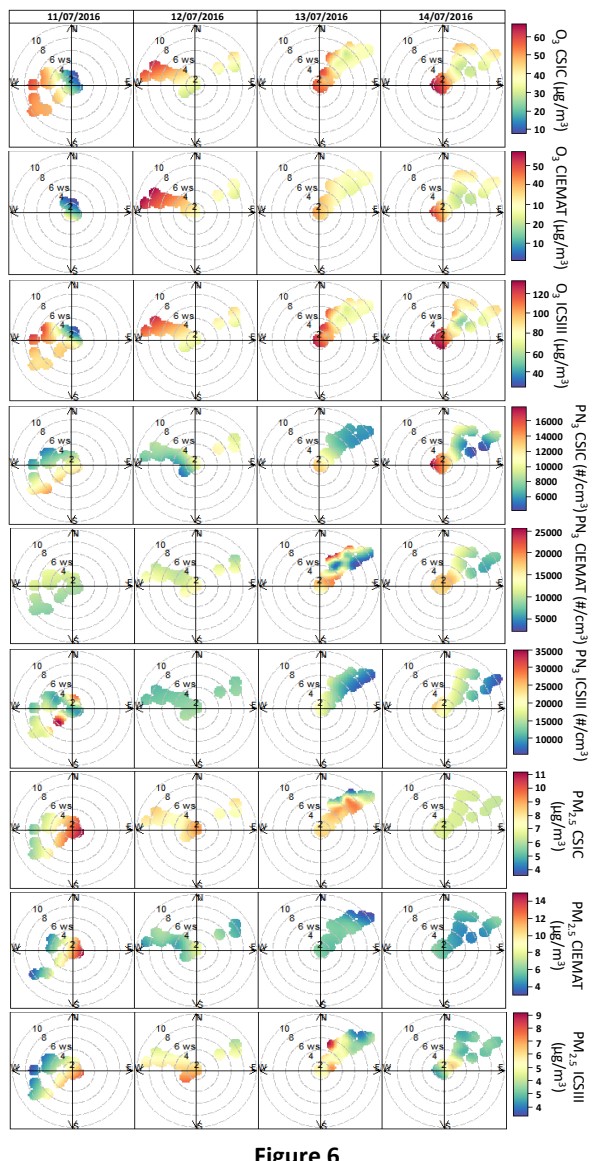

**Figure 6**





**Figure 7**


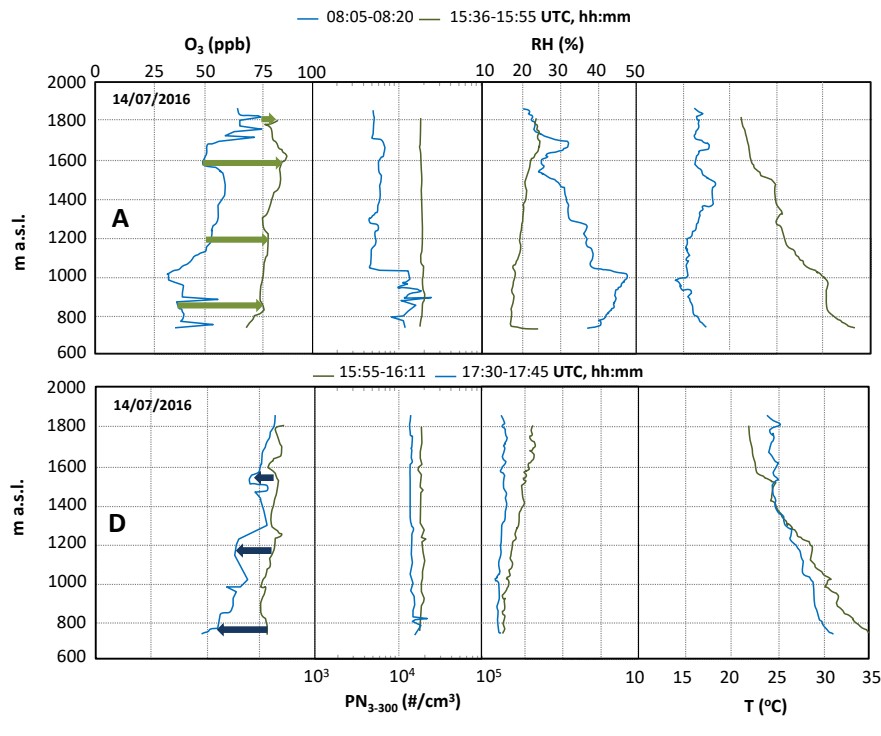


**Figure 8**





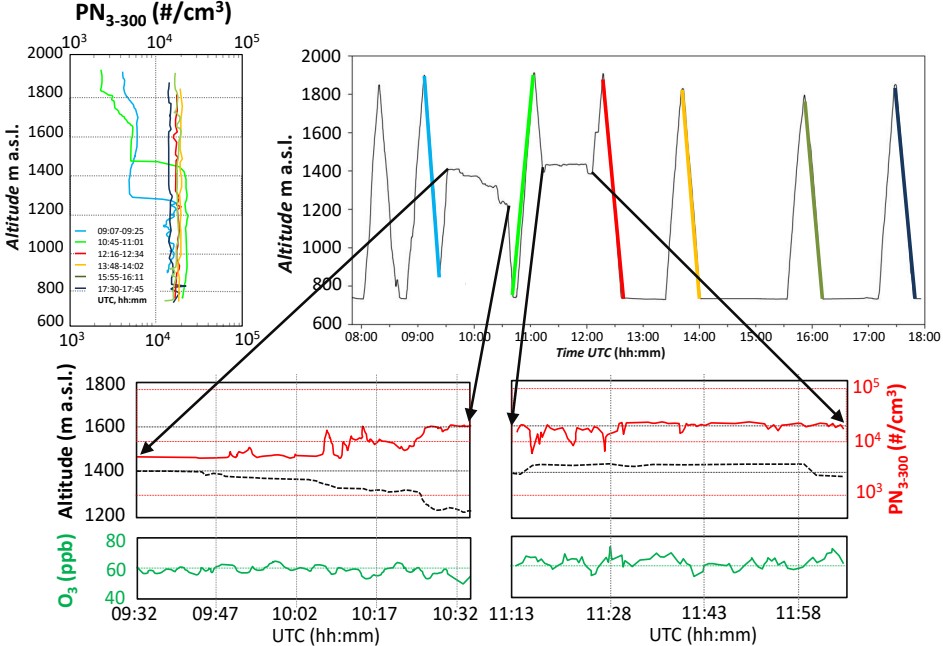

**Figure 9**




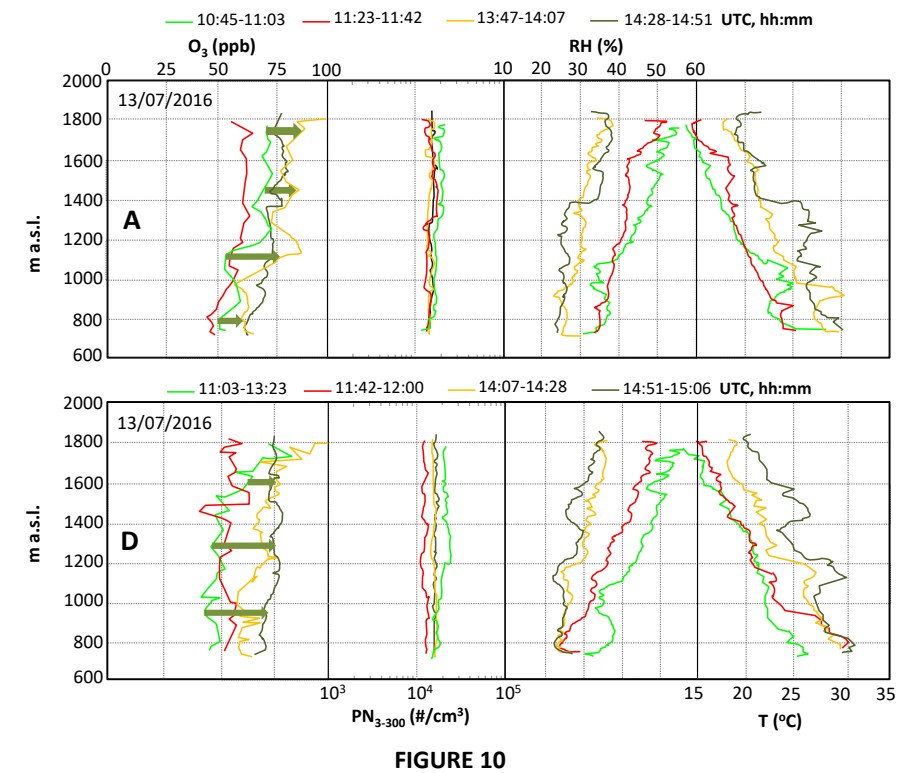

**FIGURE 10**





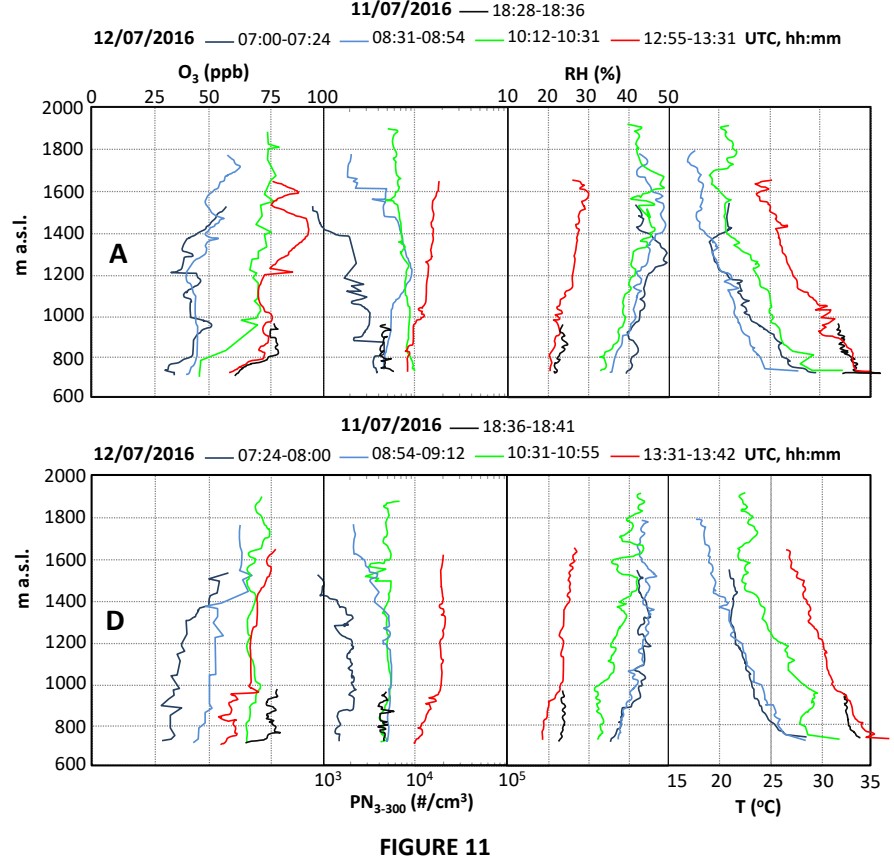

**FIGURE 11**



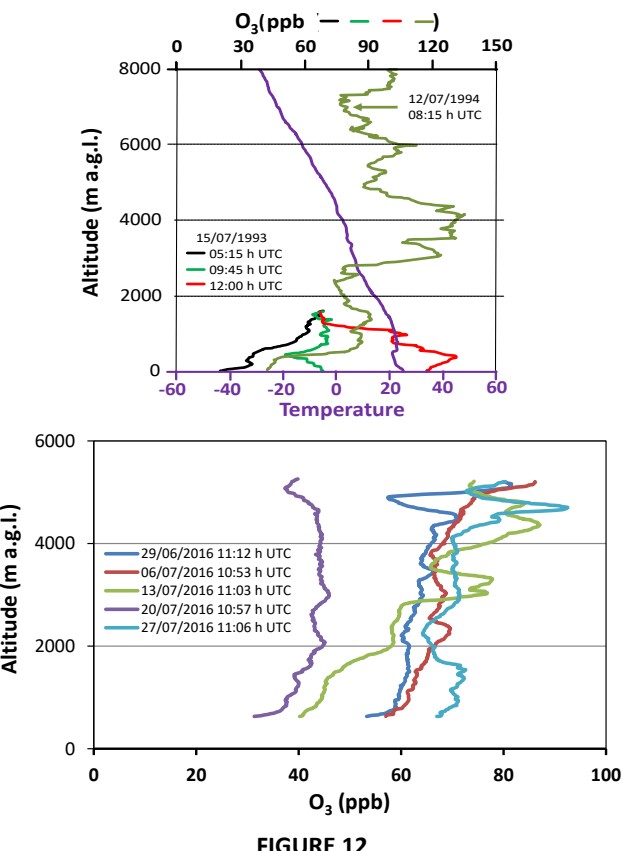


**FIGURE 12**





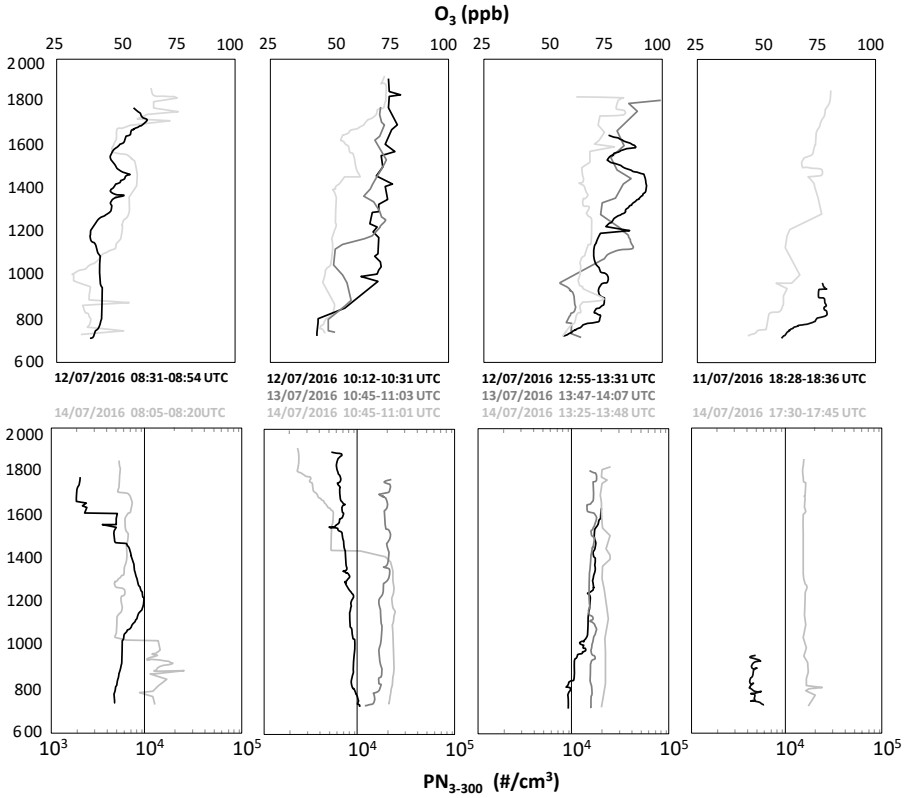

**Figure 13**





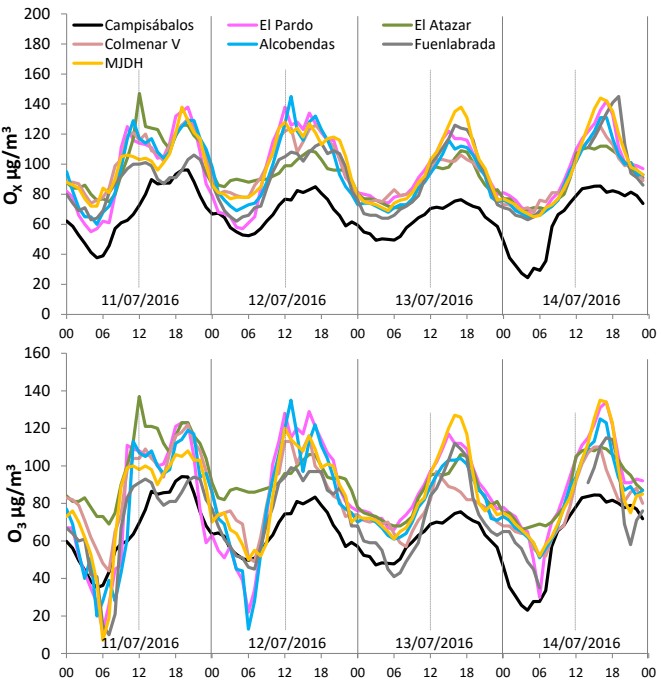

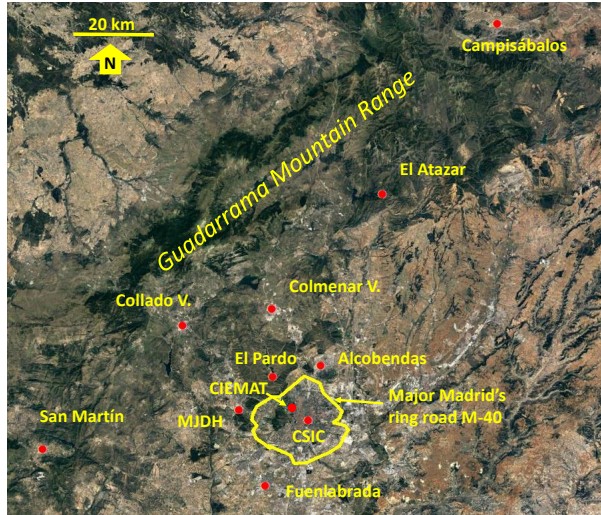

**FIGURE 14**



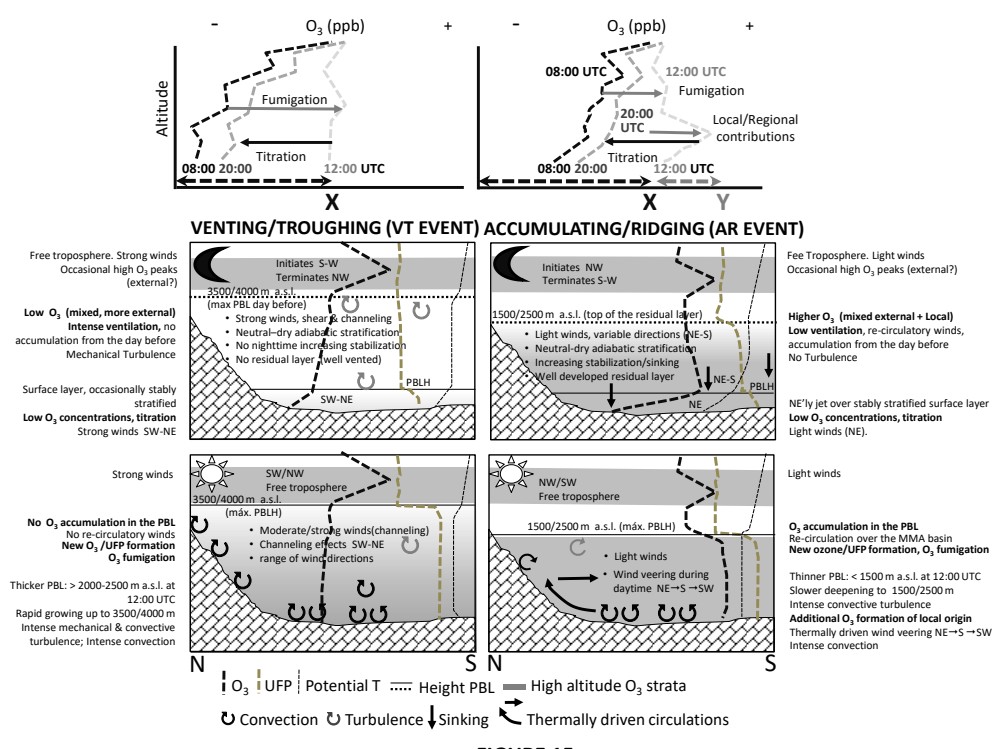


**FIGURE 15**



**TABLES**

Table 1. Details of the instrumentation used in the three supersites and the platform mounted on tethered balloons.

| Site | Latitude (N) | Longitude (W) | Altitude (m a.s.l.) | Parameter (Device-Model) | Operation period |
|---|---|---|---|---|---|
| CSIC | 40º26'25" | 03º41'17" | 713 | NOx (Teledyne API 200EU)<br>O3 (2B Technologies 202)<br>UFP>2.5nm (CPC-TSI 3775)<br>BC (Aethalometer-AE33)<br>PM1 (OPC-GRIMM 1107) | 09-20/07/2016 |
| CIEMAT | 40º27'23" | 03º43'32" | 669 | $NO_x$ (THERMO 17i)<br>$O_3$ (THERMO 49i)<br>UFP>7nm (CPC-TSI 3772)<br>UFP>2.5nm (CPC-TSI 3776)<br>BC (Aethalometer-AE33)<br>PM2.5 (TEOM©)<br>Meteorological tower | 04-20/07/2016 |
| ISCIII | 40º27'27" | 03º51'54" | 739 | $NO_x$ (THERMO 17i)<br>$O_3$ (THERMO 49i)<br>UFP>7nm (CPC-TSI 3783)<br>UFP>2.5nm (CPC-TSI 3776)<br>BC (MAAP–THERMO)<br>$PM_1$ (OPC-GRIMM 1108)<br>PTR-ToF-MS (HR 8000, Ionicon)(operating procedures described in SI) | 04-20/07/2016 |
| MJDH-RC (vertical profiles) | 40º28'30" | 03º52'55" | 729 | UFP>3nm (CPC Hy-CPC)<br>$O_3$ (PO3M$^{TM}$ 2B Technologies)<br>Meteorology (Temp., RH, Press., wind speed and direction) | 11-14/07/2016 |





Table 2. Vertical measurement profiles obtained during 11-14/07/2016 at Majadahonda (MJDH-RC).
918

| Day | Starting hour (UTC) | Final hour (UTC) | Number of profiles | Maximum height (m a.s.l.) |
|---|---|---|---|---|
| 11/07/2016 | 18:30 | 18:45 | 2 | 200 |
| 12/07/2012 | 07:02 | 07:40 | 2 | 850 |
| | 08:30 | 09:10 | 2 | 1000 |
| | 10:10 | 10:56 | 2 | 1100 |
| | 11:55 | 13:43 | 2 | 900 |
| 13/07/2008 | 10:45 | 11:25 | 2 | 1000 |
| | 11:25 | 12:00 | 2 | 1000 |
| | 13:47 | 14:29 | 2 | 1000 |
| | 14:29 | 15:12 | 2 | 1100 |
| 14/07/2004 | 08:03 | 08:44 | 2 | 1150 |
| | 08:48 | 10:37 | 2 | 1100 |
| | 10:46 | 12:45 | 2 | 1200 |
| | 13:22 | 14:02 | 2 | 1100 |
| | 15:23 | 16:13 | 2 | 1025 |
| | 17:12 | 17:31 | 2 | 1100 |

919