# Peer review of "Metropolitan Area, central Spain"

_Atmospheric Chemistry and Physics, 2017_

## Referee Comment (RC1) · Anonymous Referee #1 · 11 Jan 2018

Overview: The paper deals with the phenomenology of the summer ozone episodes over the greater area of Madrid, Spain. I think that it is a very interesting study, analyzing atmospheric measurements of ozone and fine particles together with many other atmospheric parameters and giving further insight to the complicated atmospheric mechanisms related with air pollution over the area. In my opinion, the document deserves publication in ACP, after the recommendations listed below are taken into account.

General comments: A weak point of the paper is that the levels of measured surface ozone are mainly related (or attributed) to the photochemical ozone production over the metropolitan area of Madrid. On the other hand, I think that the variations of

the background ozone levels within the boundary layer and the free troposphere are not discussed with sufficient detail. In relation to that comment and based on research results carried out on the other side of the Mediterranean basin, in the Eastern Mediterranean, it comes out that the regional background ozone levels in the free troposphere and the boundary layer during summer, regularly exceeding 60 ppb, contribute on the average to the greatest part of the surface ozone levels measured in large urban areas like Athens (Kalabokas et al., 2000; Kourtidis et al., 2002; Kouvarakis et al., 2002; Lelieveld et al., 2002; Kalabokas and Repapis, 2004; Gerasopoulos et al., 2005). The main origin of these high ozone background levels over the Eastern Mediterranean is tropospheric ozone subsidence, which seems to be strongly related with specific synoptic meteorological conditions, occurring very frequently during summer at the Eastern side of the Mediterranean basin (Kalabokas et al., 2013; Zanis et al., 2014; Kalabokas et al., 2015; Akritidis et al., 2016). In addition, recent research shows that during springtime ozone episodes (April – May) over the western Mediterranean similar synoptic meteorological patterns might also occur and which are linked with regional episodes mainly induced by large scale tropospheric ozone subsidence, influencing (or fumigating) the boundary layer as well as the ground surface ozone concentrations (Kalabokas et al., 2017).

Even if the typical meteorological conditions prevailing over the Iberian Peninsula during summer are quite different than in the Eastern Mediterranean, as it is very well described in the introduction of the manuscript, occasionally such conditions might occur. In fact, I think that this is the case of the ozone episode of 11-15 July 2016, which is the most studied period in the manuscript (when the intensive measuring campaign has taken place). As shown in Figs 3 and 12, the free tropospheric ozone levels are much higher on July 13, 2016 than the two weeks before and after and at the same time the relative humidity values in the lower troposphere are close to zero (and being in sharp contrast with the periods before and after). In fact, this feature is a very common characteristic of deep and large-scale tropospheric subsidence in summertime ozone vertical profiles over the Eastern Mediterranean, indicating an origin of air

masses from the upper tropospheric or stratospheric layers (Kalabokas et al., 2013; Kalabokas et al., 2015).

Therefore, for a better assessment of the free tropospheric influence as well as the reported fumigation events over the area, I would suggest putting more emphasis on the analysis of the synoptic conditions during this most studied period, when the intensive measurement campaign has taken place (11-15 July 2016). A figure could be added including at least the daily meteorological maps of geopotential height, omega vertical velocity and specific humidity at 700hPa pressure level (representative for the free tropposphere), which I think that they would be sufficient to follow satisfactorily the evolution and the geographical extent of the subsidence phenomenon (the subsiding air mass seems to originate from N-NW Atlantic). If this parameter is taken into account, then I think that the discussion concerning the origin of the fumigation events during 11-16 July 2016 would be more complete (tropospheric ozone subsidence in addition to the local ozone photochemical production associated with valley-breeze recirculation and ozone residual layers, as mentioned many times in the manuscript). So, I would suggest modifying accordingly the respective paragraphs, where sometimes the high ozone values recorded at the top of the boundary layer are not fully explained (e.g.: Page 5, lines 175-187; Page 9, lines 348 – 352, 362-365; Page 12, lines 470 – 473, 483-487; Page 14, lines 570 – 582).

In addition, I think that it would be more appropriate to refer to "Western Mediterranean" ozone when analyzing ozone in Spain (instead of "Mediterranean" or "S. Europe" in general) as, according to the above mentioned papers, the phenomenology of the summertime ozone over the Eastern Mediterranean seems to be is quite different than the typical ozone phenomenology over the Western Mediterranean.

I would strongly recommend taking these considerations into account, which have been made in the spirit to further improve this good quality manuscript, by modifying the respective paragraphs. After responding to these remarks, I think that the paper is ready for publication.

Technical comments: Fig. 4: Very condensed and difficult to follow, especially on printed paper. I would suggest splitting into two parts and eventually using gridlines.

---

## Short Comment (SC1) · 11 Jan 2018

Dear referee#1,

Thanks a lot for your review and constructive critical comments on our paper. We fully agree with your commments on the contribution of subsidence O3 that, as you commented, we mentioned in the introductory section but not considered in an appropriate way when interpreting surface data. For sure we will follow your suggestions in addressing these changes in the revised version.

We will refer also to W Mediterranean instead of Mediterranean or S Europe, when

referring to the spatial ambit of application for our results for O3.

Thanks a lot again for your co0nstructive comments

Xavier Querol

---

## Referee Comment (RC2) · Anonymous Referee #2 · 8 Feb 2018

The study aimed to explore if changes in nucleation of ultrafine particles and high surface ozone levels in urban areas could be related to the atmospheric dynamics changes over the European region at highest ozone risk. The content and the methodology are consistent and this study reports valuable conclusions based on experimental investigation by means of simultaneous vertical profiles of concentrations for both pollutants and meteorological parameters using balloons.

However, the conclusions can be improved (other than ozone levels are low with thick PBL and high wind speed) and some parts are really heavy going and need to be reformulated. In general, the English language can be improved and some parts need

to be reformulated in a simpler way. The entire paper needs to be (deeply) reorganized (i.e. section Results & Discussions & Materials and Methods) and shortened. The term "altitude" must be correctly used, i.e. use "altitude" for a.g.l. (for vertical profiles) and "elevation" for a.s.l. Please, revise all figures and manuscript.

Abstract - Line 34: "high O3 level" or "ground-level O3"

Introduction - The state-of-the-art is consistent, however for a non-European scientist; the references are limited to Spain instead of Western Mediterranean region. The state-of-the-art must be documented e.g. including an overview of ozone impacts (Ochoa-Hueso et al., 2017, EnPo) and I found different studies (Lelieveld et al., 2002_Science; Kalabokas et al., 2008_ Atm Env; Giannakopoulos et al., 2009_Global&Planetary Change; Velchev et al., 2011_ACP; Sicard et al., 2013_Atm Env).

Particulate matter and tropospheric ozone are the most threatening air pollutants in cities (EEA, 2015). More than 75% of the urban population is exposed to levels exceeding WHO guidelines for PM2.5, PM10 an surface O3 (EEA, 2015).

Line 66: "implemented" by? Line 70: add a reference. Line 73: add an example for "easy to identify". Lines 76-77, add a reference, what are the climatic and geographical characteristics leading to high ozone levels? In summer, the western part of the Mediterranean basin is dominated by anti-cyclonic subsidence (Hadley circulation + Azores), high pressure, low winds and strong insolation and thus atmospheric stability favoring photochemical production of ozone (Kalabokas et al., 2008; Giannakopoulos et al., 2009; Velchev et al., 2011; Sicard et al., 2013) and emissions of biogenic VOCs (Giannakopoulos et al., 2009). Line 90: add a value and a reference for "peaks O3 concentrations". The ozone control measures are effective at rural sites while ozone levels are rising in cities (Paoletti et al., 2014_EnPo; Sicard et al., 2013_Atm Env). Line 99: NO titration, add a reference. What is the situation in Spain as compared to other European countries such as France and Italy? Line 101: the reduction in NOx

and VOC emissions within the EU started in the early 1990s (not late). Line 107: the urban areas are characterized by "VOC-limited" conditions, and a reduction in NOx emission increases the O3 formation. Lines 112-116: too many references and auto-citation, the state-of-the-art need to be enlarged (e.g. Italy, France, Portugal) and add a comparison + quantification of the spatio-temporal changes for surface ozone levels. Lines 124-127: to be reformulated. Line 128: what are the scenarios? Line 139: "low pre-existing. . .", please explain. Line 142: add the period of the study.

Materials & methods - The experimental protocol and methodology are consistent, well described. Even if this section is important, it can be shortened, e.g. lines 165-190: move to section "Discussion" and need to be shortened. Line 195: "highest levels of O3", how much? Line 218: "very good results", please quantify (e.g. $r^2$) this statement.

Results – Please avoid describing Figures (e.g. 260-262, 318-320), this is really heavy. Lines 249-253: move to section "Materials & Methods". Line 254: what is the correlation coefficient between NO2 and ozone? Lines 271-272: move to section "Discussion". Lines 260-264: move to section "Materials & Methods". Lines 271-272: move to section "Discussion". Line 344: "high hourly O3 concentrations. . ."Lines 345-348: move to section "Discussion". Lines 353-414: long section, boring, need to be shortened.

Discussion – Line 425: "the surface O3 concentration". Line 441: "biogenic VOICs", please specify the source (e.g. isoprene). Lines 459-464: move to section "Results". Line 468: titration by NO. Lines 487-495: to be reformulated, unclear. Lines 500-505: typical diurnal ozone concentrations, well documented in the literature, not innovative for ACP. Lines 506-5014: move to section "Results". Lines 518-519: confusing compared to line 560, need to be reformulated. Lines 538-541: move to section "Results".

At high elevation, changes in the background tropospheric ozone can be attributed to to i) hemispheric background concentrations, in part due to the reduction in NOx emissions; ii) the exchange between the free troposphere and the boundary layer and iii) the stratospheric inputs (Chevalier et al., 2007_ACP, Kulkarni et al., 2011; Lefohn et

al., 2012_Atm Env; Sicard et al., 2016_Env Research). The "stratospheric inputs" need to be discussed in this section, as well as exchanged between the lower stratosphere and free troposphere.

Conclusions – Line 587-588: climate change might reduce the benefits of the ozone control strategies. This can be discussed e.g. climate change and the measures and policies in North America or Asia will need to be considered into future ozone policies in Europe for ozone mitigation (Lefohn and Cooper, 2015_Atmo Env; Sicard et al., 2017_ACP). Line 612: a statement of ACP Special Issues (Lamarque et al., 2013_ACP and Young et al. 2013_ACP) with respect to the ACCMIP models can be done, or the validation of ACCMIP ozone simulations using sonde throughout the free troposphere and lower stratosphere for both seasonal and year-to-year variations of ozone (Kazuyuki Miyazaki and Kevin Bowman, 2017_ACP).

Grammatical suggestions and typos

Lines 69, 88: PM 10 and 2.5 in subscript Line 86: Monks et al., 2015 rather than 2014. Line 133: to be reformulated as "recorded in July" (a simpler way) Line 142: why did you put (and UFP) with brackets? Line 188: Gómez-Moreno (hyphen) Some acronyms need to be defined e.g. lines 206, 215, 236 Line 206: 19 or 20 July, to be checked Line 442: "through the" (space) Line 444: remove all the text given in parentheses (really heavy) Line 475: McKendry and Lundgren (instead of "et al.") Line 601: add a reference

Tables & Figures – There are too many figures, it would be necessary to select the most informative figures (5-6 maximum) and move the other to SI. Please, change "altitude" as "elevation" in Figures and Tables when necessary. Figures 7-10-11 can be joined. Too many Figures seem similar and not useful. Figure 1: put units (m) on X- and Y-axis and "elevation (a.s.l). Figure 3: how is possible to read variations of values, I don't see the units for each parameter (e.g. T, RH), please add a second Y-axis with units. Figure 4: blurred. Figure 13: units on Y-axis. Table 1: add the station "El Retiro" and define

the acronyms in caption (RH, Temp., UFP. . .). Table 2: m (a.g.l).

References - Kulmala et al., 2000 (Line 702), Millán & Artíñano, 1992 (Line 725) & Skrabalova et al., 2015 are missing. Line 766: Pujadas et al., the year is missing (2000) Line 781: "et al.," please supply the full author list. Please, read the advices in the guidelines for authors for the "list of references" (e.g. publication year, lines 787, 702) and consider a chronological order for the same author e.g. Millán.

---

## Author Comment (AC1) · 19 Feb 2018

Dear anonymous referee #1,

Thanks a lot for considering our manuscript suitable for publication after revision and also for the comments and suggestion that you gave us, especially those on the stratospheric $O_3$, which improved a lot the article. As you will see in the revised version we took into account all your valuable comments and suggestions. We also attach below a report on changes addressed and reply to your specific queries.

Following one of your suggestions, to evaluate the stratospheric episode, we requested the support of Prof. M. Millán, who accepted and contributed a lot; so we have included him as a co-author in this revised version.

Thanks a lot for your constructive review.

Xavier Querol on behalf of all co-authors

**Report on changes addressed and reply to specific queries**

*Overview: The paper deals with the phenomenology of the summer ozone episodes over the greater area of Madrid, Spain. I think that it is a very interesting study, analysing atmospheric measurements of ozone and fine particles together with many other atmospheric parameters and giving further insight to the complicated atmospheric mechanisms related with air pollution over the area. In my opinion, the document deserves publication in ACP, after the recommendations listed below are taken into account.* REPLY, Thanks a lot for considering our manuscript suitable for publication after revision and also for the comments and suggestion on the stratospheric $O_3$ that improved a lot the article.

*General comments: A weak point of the paper is that the levels of measured surface ozone are mainly related (or attributed) to the photochemical ozone production over the metropolitan area of Madrid.* REPLY, Thanks a lot this comment. We also considered long range transport (named 'external' in the paper. But as you might see in the revised version we have enlarged this and also the stratospheric issue that we miss in our original submission.

*On the other hand, I think that the variations of the background ozone levels within the boundary layer and the free troposphere are not discussed with sufficient detail. In relation to that comment and based on research results carried out on the other side of the Mediterranean basin, in the Eastern Mediterranean, it comes out that the regional background ozone levels in the free troposphere and the boundary layer during summer, regularly exceeding 60 ppb, contribute on the average to the greatest part of the surface ozone levels measured in large urban areas like Athens (Kalabokas et al., 2000; Kourtidis et al., 2002; Kouvarakis et al., 2002; Lelieveld et al., 2002; Kalabokas and Repapis, 2004; Gerasopoulos et al., 2005). The main origin of these high ozone background levels over the Eastern Mediterranean is tropospheric ozone subsidence, which seems to be strongly related with specific synoptic meteorological conditions, occurring very frequently during summer at the Eastern side of the Mediterranean basin (Kalabokas et al., 2013; Zanis et al., 2014; Kalabokas et al., 2015; Akritidis et al., 2016). In addition, recent research shows that during springtime ozone episodes (April – May) over the western Mediterranean similar synoptic meteorological patterns might also occur and which are linked with regional episodes mainly induced by large scale tropospheric ozone subsidence, influencing (or fumigating) the boundary layer as well as the ground surface ozone concentrations (Kalabokas et al., 2017).*
REPLY, Thanks a lot for these observations. Yes you are completely right; we miss these issues in our discussions and interpretations. We have added these observations and discussions in several part of the revised paper. For example text added or changed includes:

- Abstract: We added: The results demonstrate the concatenation of venting and accumulation episodes, with relative $O_3$ lows (venting) and peaks (accumulation) in surface levels. Regardless of the episode type, fumigation of high altitude $O_3$ (from different origins) contributes the major proportion of surface $O_3$ concentrations.
- Section 1: New text added: In the Eastern Mediterranean, the regional background O3 levels in the free troposphere and the boundary layer during summer might regularly exceed 60 ppb, and fumigation of these upper air masses contribute on the average to the greatest part of the surface O3 levels measured in Greece (Kalabokas et al., 2000; Kourtidis et al., 2002; Kouvarakis et al., 2002; Lelieveld et al., 2002; Kalabokas and Repapis, 2004; Gerasopoulos et al., 2005). Furthermore, a number of studies reported contributions of stratosphere to the surface O3 concentrations during specific

meteorological scenarios in the same region (Kalabokas et al., 2013, 2015; Zanis et al., 2014; Parrish et al., 2012; Lefohn et al. , 2012; Akritidis et al., 2016, among others). In addition, recent research shows that during springtime O3 episodes (April – May) over the WMB similar synoptic meteorological patterns might also occur; and that these are linked with regional episodes mainly induced by large scale tropospheric O3 subsidence, influencing the boundary layer as well as the ground surface O3 concentrations (Kalabokas et al., 2017). However, the most intense episodes in the WMB occur in June-July according to the statistics for the 2000-2015 period by Querol et al. (2016) for Spain.

- Section 3.1. New text added: The AEMET free-sounding shows low O3 surface concentrations (<45 ppb) and high levels (>70 ppb) in the middle troposphere (3000-5000 m a.s.l.), associated with very low relative humidity and intense W to NW winds blowing at that height, which will be discussed in section 4.
- Section 4: discussion added, see below.
- New figure added (see below)
- Conclusions modified (see below)

*Even if the typical meteorological conditions prevailing over the Iberian Peninsula during summer are quite different than in the Eastern Mediterranean, as it is very well described in the introduction of the manuscript, occasionally such conditions might occur. In fact, I think that this is the case of the ozone episode of 11-15 July 2016, which is the most studied period in the manuscript (when the intensive measuring campaign has taken place). As shown in Figs 3 and 12, the free tropospheric ozone levels are much higher on July 13, 2016 than the two weeks before and after and at the same time the relative humidity values in the lower troposphere are close to zero (and being in sharp contrast with the periods before and after). In fact, this feature is a very common characteristic of deep and large-scale tropospheric subsidence in summertime ozone vertical profiles over the Eastern Mediterranean, indicating an origin of air masses from the upper tropospheric or stratospheric layers (Kalabokas et al., 2013; Kalabokas et al., 2015).*

*Therefore, for a better assessment of the free tropospheric influence as well as the reported fumigation events over the area, I would suggest putting more emphasis on the analysis of the synoptic conditions during this most studied period, when the intensive measurement campaign has taken place (11-15 July 2016). A figure could be added including at least the daily meteorological maps of geopotential height, omega vertical velocity and specific humidity at 700hPa pressure level (representative for the free troposphere), which I think that they would be sufficient to follow satisfactorily the evolution and the geographical extent of the subsidence phenomenon (the subsiding air mass seems to originate from N-NW Atlantic). If this parameter is taken into account, then I think that the discussion concerning the origin of the fumigation events during 11-16 July 2016 would be more complete (tropospheric ozone subsidence in addition to the local ozone photochemical production associated with valley-breeze recirculation and ozone residual layers, as mentioned many times in the manuscript). So, I would suggest modifying accordingly the respective paragraphs, where sometimes the high ozone values recorded at the top of the boundary layer are not fully explained (e.g.: Page 5, lines 175-187; Page 9, lines 348 – 352, 362-365; Page 12, lines 470 – 473, 483-487; Page 14, lines 570 – 582).*

REPLY. Thanks a lot for these very interesting comments. Indeed we have applied all your suggestions and modified the introduction, discussion and conclusions sections to this end. Examples are:

- Section 4. We added the following discussion: Considering the free sounding O3 profiles in Figure 3, high O3 concentrations (>70 ppb) can be observed above the PBL, between 3000 and 5000 m a.s.l., which may be related to larger scale transport of pollutants, previously

uplifted to the mid-troposphere or originated after a stratospheric intrusion and a subsequent deep subsidence into the middle troposphere, as it is probably the case based on the ECMWF ERA-Interim reanalysis data. Transport of high O3 air masses in the middle troposphere, as for the 13/07/2016 in Figure 3, was also documented by Plaza et al. (1997) over this area in July 1994, during the final phase of a high O3 period. More recently, Kalabokas et al. (2013, 2015, 2017), Zanis et al. (2014) and Akritidis et al (2016), among others, have shown that similar transport processes of enriched O3 layers at high altitude, can contribute to increase surface O3 concentrations during the summer in the Eastern Mediterranean. This transport has been associated with large scale subsidence within strong northerly winds in the Eastern Mediterranean (Etesian winds), and the affected layers are dryer than average and show negative temperature anomalies. Figure S11 shows the ECMWF ERA-Interim reanalysis together with the AEMET O3 free soundings at Madrid airport for the 13/07/2016. The ridging at the lower troposphere over the Bay of Biscay at the rear of an upper-level trough (left panels) is accompanied by intense NW winds blowing at the middle and upper troposphere and NE winds at ground level and up to 2000 m (see the radiosonde profile in the same figure). The O3 intrusion is associated to the upper-level trough (Sections A-A and B-B in the figure), and a large area of deep subsidence and extremely low relative humidity observed within the NW flows over Madrid and to the north of the Iberian Peninsula and the Bay of Biscay. High O3 concentrations values and low relative humidity of the ERA-Interim profiles over the airport of Madrid (green and red dotted-lines in the panel "g" of the Figure S11) are in agreement with the radiosonde observations in the same panel.

The question now is how much of this O3 could fumigate at ground level. According to the radiosonde data, the mixing height top was about 2000 m a.s.l. at midday, but could increase to about 3100 m a.s.l. after the projection of the surface temperature increase observed during the afternoon at near-by stations. This height reaches the lower part of the O3 enriched layer originated in the tropopause folding. Thus, a certain impact seems likely. However, the O3 concentrations were relatively low at all surface stations during that day, as it corresponds to a vented low O3 period.

- A new figure S11 has been added and discussed to show the stratospheric intrusion following your suggestion.
- We added these two paragraphs in the conclusions:
- The O3 source apportionment is very complex, having contributions from local/regional and remote sources, including the stratosphere. The relative contributions of these might vary in time and space (e.g. Lefohn et al., 2014).

- Climate change my reduce the benefits of the $O_3$ abatement policy (since hot waves increase $O_3$ episodes), and this, as well as the measures and policies in N America and Asia will need to be considered into future Europe policies for $O_3$ mitigation (Lefohn and Cooper, 2015; Sicard et al., 2017).

- And we modified the fourth as follows: ·        ·In the MAB, during the highest O3 (accumulation) episodes, in addition of the contribution (to surface concentrations) by fumigation of upper O3 (from regional transport, hemispheric free troposphere O3, and intruded stratospheric O3, X in Figure 10), there is an added fraction produced locally and transported-recirculated within the MAB, which accumulates from one day to the next (Y in Figure 10).

*In addition, I think that it would be more appropriate to refer to "Western Mediterranean" ozone when analyzing ozone in Spain (instead of "Mediterranean" or "S. Europe" in general) as, according to the above mentioned papers, the phenomenology of the summertime ozone over the Eastern Mediterranean seems to be is quite different than the typical ozone phenomenology over the Western Mediterranean.*
REPLY: Yes we have revised and changed following your suggestion. Thanks for this.

*I would strongly recommend taking these considerations into account, which have been made in the spirit to further improve this good quality manuscript, by modifying the respective paragraphs. After responding to these remarks, I think that the paper is ready for publication.*
REPLY: Yes we did implement your very good observations and we thank you a lot for your very valuable review.

*Technical comments: Fig. 4: Very condensed and difficult to follow, especially on printed paper. I would suggest splitting into two parts and eventually using gridlines.*
REPLY: We made it again with clearer drawings.

END OF THE REPORT

---

## Author Comment (AC2) · 19 Feb 2018

Dear anonymous referee #2,

Thanks a lot for considering our manuscript suitable for publication after revision and also for the comments and suggestion that you gave us, which improved a lot the article. As you will see in the revised version we took into account all your valuable comments and suggestions. We also attach below a report on changes addressed and reply to your specific queries.

Following your suggestions we also evaluated in detail the stratospheric contributions. To this end we requested the support of Prof. M. Millán, who accepted and contributed a lot; so we have included him as a co-author in this revised version.

Thanks a lot for your constructive review.

Xavier Querol on behalf of all co-authors

**Report on changes addressed and reply to specific queries**

*The study aimed to explore if changes in nucleation of ultrafine particles and high Surface ozone levels in urban areas could be related to the atmospheric dynamics changes over the European region at highest ozone risk. The content and the methodology are consistent and this study reports valuable conclusions based on experimental investigation by means of simultaneous vertical profiles of concentrations for both pollutants and meteorological parameters using balloons.*

*However, the conclusions can be improved (other than ozone levels are low with thick PBL and high wind speed) and some parts are really heavy going and need to be reformulated. In general, the English language can be improved and some parts need to be reformulated in a simpler way. The entire paper needs to be (deeply) reorganized (i.e. section Results & Discussions & Materials and Methods) and shortened.*

REPLY, Thanks a lot for considering our manuscript suitable for publication after revision and also for the detailed revision done that improved a lot the presentation of our results. We have also revised again English usage.

*The term "altitude" must be correctly used, i.e. use "altitude" for a.g.l. (for vertical profiles) and "elevation" for a.s.l. Please, revise all figures and manuscript.* REPLY: Corrected in all text, figures and tables.

*Abstract - Line 34: "high O3 level" or "ground-level O3".* REPLY: Done.

*Introduction - The state-of-the-art is consistent, however for a non-European scientist; the references are limited to Spain instead of Western Mediterranean region. The state-of-the-art must be documented e.g. including an overview of ozone impacts (Ochoa-Hueso et al., 2017, EnPo) and I found different studies (Lelieveld et al., 2002_Science; Kalabokas et al., 2008_ Atm Env; Giannakopoulos et al., 2009_Global&Planetary Change; Velchev et al., 2011_ACP; Sicard et al., 2013_Atm Env).*

*REPLY: Thanks for the constructive comment. Yes indeed we went directly to focus referencing of Millan's team in Spain, because they were pioneering the ozone research in this region in the 1980s-1990s, and in our opinion the built the basis of knowledge for atmospheric dynamics governing O3 levels in The Western Mediterranean, clearly different from those prevailing in the Eastern side of the basin. We recognize now that we missed first reviewing the whole Mediterranean issue. We have added now this view and included references you supplied us with and commented them in the introduction and discussion sections.*

*Particulate matter and tropospheric ozone are the most threatening air pollutants in cities (EEA, 2015). More than 75% of the urban population is exposed to levels exceeding WHO guidelines for PM2.5, PM10 an surface O3 (EEA, 2015).* REPLY, we understand that you require us to add this into our text, and we did it using updated EEA (2017). Thanks

*Line 66: "implemented" by*? REPLY: Added: by the national, regional and local administrations

*Line 70: add a reference.* REPLY Added: EEA, 2017

*Line 73: add an example for "easy to identify".* REPLY: Added: (such as abating industrial, shipping and traffic emission with catalytic converters for $NO_X$ and particulate controls for PM).

*Lines 76-77, add a reference, what are the climatic and geographical characteristics leading to high ozone levels? In summer, the western part of the Mediterranean basin is dominated by anti-cyclonic subsidence (Hadley circulation +Azores), high pressure, low winds and strong insolation and thus atmospheric stability favouring photochemical production of O₃ (Kalabokas et al., 2008; Giannakopoulos et al., 2009; Velchev et al., 2011; Sicard et al., 2013) and emissions of biogenic VOCs (Giannakopoulos et al., 2009).* REPLY: Added: In summer, the Western Mediterranean Basin (WMB), surrounded by high mountains, falls under the influence of the semi-permanent Azores anticyclone. Clear skies prevail under a generalized level of subsidence aloft, and meso-meteorological processes with marked diurnal cycles dominate. Re-circulations, strong insolation and stability of upper layers favour the production/accumulation of O3 (Millán et al., 1997, 2000 and 2002; Kalabokas et al., 2008; Giannakopoulos et al., 2009; Velchev et al., 2011; Sicard et al., 2013), and the emissions of biogenic volatile organic compounds (BVOCs) (Giannakopoulos et al., 2009).

*Line 90: add a value and a reference for "peaks O3 concentrations".* REPLY: We added (those exceeding the hourly information threshold of 180 µg/m$^3$)

The ozone control measures are effective at rural sites while ozone levels are rising in cities (Paoletti et al., 2014_EnPo; Sicard et al., 2013_Atm Env). REPLY: Thank you, yes we added these references to the others that find this trend in Europe. We missed them and these are very important.

*Line 99: NO titration, add a reference. What is the situation in Spain as compared to other European countries such as France and Italy?* REPLY: Yes we added reference; This NO/NO2 is widely accepted in Europe. We added: This trend to decrease the NO/NO2 rate from diesel vehicle emissions (the main source of NOx in urban Europe) has been widely reported (i.e. Carslaw et al., 2016)

*Line 107: the urban areas are characterized by "VOC-limited" conditions, and a reduction in NOx emission increases the O3 formation.* REPLY, Yes we also gave this possibility in the original paper, but to make it more clear we added: We still do not know if this increase is due to a decrease in the NO titration effect or to the fact that the O3 formation is by VOCs dominated, since the urban areas are characterized by 'VOC-limited' conditions, and a reduction in NOx emission might yield to an increase of the O3 formation.

 *Lines 112-116: too many references and autocitation, the state-of-the-art need to be enlarged (e.g. Italy, France, Portugal) and add a comparison + quantification of the spatio-temporal changes for surface ozone levels.* REPLY. OK we changed this focus on the whole Mediterranean and we added the references (but no auto-citation, no one paper of the main author of this paper is cited here). Now we stated: Intensive research on O3 pollution has been carried out in the Mediterranean since the late 1980s, which has been key to understand the behaviour of this pollutant in Europe, and to establish the current air quality European standards (Millán et al., 1991, 1996a, 1996b, 1996c, 2000, 2002; Millán, 2002; Lelieveld 2002; EC, 2002, 2004; Millán and Sanz, 1999; Mantilla et al., 1997; Salvador et al., 1997, 1999; Gangoiti et al., 2001; Stein et al., 2004, 2005; Chevalier et al. , 2007; Kalabokas et al., 2008, 2015 and 2017; Castell et al., 2008a,

2008b, 2012; Kulkarni et al., 2011; Velchev et al., 2011; Doval et al., 2012; Sicard et al., 2013; Millán et al., 2014; Escudero et al., 2014; Zanis et al., 2014; Sicard et al., 2017; among others). EEA (2017) reports a clear gradient to increase the exceedances of the human protection 8-h O3 target value in Southern and Central Europe, higher in the Italian Po Valley and Spain, and relatively lower in Portugal and the Eastern Mediterranean.

*Lines 124-127: to be reformulated.* REPLY: Done: Querol et al. (2017) reported that the high-O3 plume transported from the metropolitan area of Barcelona contributed decisively to cause the frequent exceedances of the information threshold in the northern areas of Barcelona during the acute O3 episodes in July 2015.

*Line 128: what are the scenarios?* REPLY: We modified the text: They also demonstrated that the meteorology associated was very complex, similar to the scenarios of vertical recirculation of air masses reported by………….

*Line 139: "lowpre-existing: : :", please explain.* *REPLY: Modified to: ….* low condensation sink potential (i.e. relatively clean atmospheres with low surface aerosol concentrations).

*Line 142: add the period of the study.* REPLY: Added.

*Materials & methods - The experimental protocol and methodology are consistent, well described. Even if this section is important, it can be shortened, e.g. lines 165-190: move to section "Discussion" and need to be shortened.* REPLY: Done. We still believe that the right place was the first description of the study area, but following requirements of the referee we moved to 'discussion' and shortened.

*Line 195: "highest levels of O3", how much?* REPLY: Added (with hourly maxima sporadically exceeding 180 µg/m$^3$)

*Line 218: "very good results", please quantify (e.g. r2) this statement.* REPLY: Added to text (with $R^2$ reaching 0.65-0.98 and slopes 0.87-1.23, Minguillón et al., 2015).
*Results – Please avoid describing Figures (e.g. 260-262, 318-320), this is really heavy.* REPLY: Deleted in both cases.
*Lines 249-253: move to section "Materials & Methods".* Reply: Done

*Line 254: what is the correlation coefficient between NO2 and ozone?* REPLY, below you have it for 2 of the sites having the highest O3 (NO2 is in Y, and O3 in X),

[Figure]

*Lines 271-272: move to section "Discussion".* REPLY: We do not understand here what you mean, moving 2 lines breaks the explanation of this paragraph. We decided to keep them in.
*Lines 260-264: move to section "Materials & Methods".* REPLY: In a prior observation you suggested deleting this description and we did it. Now is only 2 lines well inserted in this section.

*Line 344: "high hourly O3 concentrations: ::"* REPLY: Corrected and added, thanks
*Lines 345-348: move to section "Discussion".* REPLY: Moved

*Lines 353-414: long section, boring, need to be shortened.* REPLY We reduced it by 60% by moving descriptions of the first 3 days of soundings to SI and synthesising the results in the main text.

*Line 441: "biogenic VOCs", please specify the source (e.g. isoprene).* REPLY: Added: The influence of aromatic VOCs on OFP rapidly decreases while the influence of biogenic VOCs (mostly isoprene followed by monoterpenes, as primary ones, and methacrolein, methyl-vinyl-ketone, isoprene-derived isomers of unsaturated hydroxy hydroperoxides (ISOPOOH) and methylglyoxal, as the main secondary species)…

Lines 459-464: move to section "Results". REPLY. This is one of the only two cases that we decided to keep the original text here because, this discussion is summarising the evolution of the UFP and O3 profiles across the day, If we move these 5 lines, we start description from midday here and we lose the early to midday scenarios.

*Line 468: titration by NO.* REPLY: Added, thanks

*Lines 487-495: to be reformulated, unclear.* REPLY: Re-written: During the whole month of July 2016 there was a clear veering of the urban plume from Madrid, with night plume transport towards SW (MJDH-San Martin de V., Figures 9 and S12), and towards NW, N-NE, and, in some cases, E-SE during the morning and midday, followed by the decoupling and onset of the evening and nocturnal flow towards SW. This veering seems to be causally associated with the high O3 levels recorded in the W to E areas surrounding northern Madrid, since the peak concentrations recorded by the official air quality network follow this spatial and temporal evolution (Figure S12) for the exceedances of the O3 information threshold.

*Lines 500-505: typical diurnal ozone concentrations, well documented in the literature, not innovative for ACP.* REPLY: Done, we delete it.

*Lines 506-5014: move to section "Results".* REPLY. This is one of the two cases that we decided to keep the original text here because, this discussion is summarising the evolution of the UFP and O3 profiles across the day.

*Lines 518-519: confusing compared to line 560, need to be reformulated. REPLY, thanks a lot, changed to:* Conversely, the lower development of the PBL on 14/07/2016 causing les surface UPF dilution and lower top-down contributions to $O_3$ to surface concentrations accounted for opposite $O_3$ and UFP the profiles.

*Lines 538-541: move to section "Results". At high elevation, changes in the background tropospheric ozone can be attributed to to i) hemispheric background concentrations, in part due*

*to the reduction in NOx emissions; ii) the exchange between the free troposphere and the boundary layer and iii) the stratospheric inputs (Chevalier et al., 2007_ACP, Kulkarni et al., 2011; Lefohn et al., 2012_Atm Env; Sicard et al., 2016_Env Research). The "stratospheric inputs" need to be discussed in this section, as well as exchanged between the lower stratosphere and free troposphere.* REPLY: Yes this was a weakness, thanks a lot. We added: In addition of the local O3, the background contribution can be also very relevant. At high elevation, changes in the background tropospheric O3 can be attributed to to (i) hemispheric background concentrations; (ii) exchange between the free troposphere and the boundary layer; and iii) stratospheric inputs (Chevalier et al., 2007; Kulkarni et al., 2011; Parrish et al., 2012, Lefohn et al., 2012; Kalabokas et al., 2013, 2015 and 2017; Zanis et al., 2014; Akritidis et al., 2016; Sicard et al., 2017).

We also modified several parts to discuss these free-troposphere and stratospheric contributions. Examples are:

- Section 4. We added the following discussion: Considering the free sounding O3 profiles in Figure 3, high O3 concentrations (>70 ppb) can be observed above the PBL, between 3000 and 5000 m a.s.l., which may be related to larger scale transport of pollutants, previously uplifted to the mid-troposphere or originated after a stratospheric intrusion and a subsequent deep subsidence into the middle troposphere, as it is probably the case based on the ECMWF ERA-Interim reanalysis data. Transport of high O3 air masses in the middle troposphere, as for the 13/07/2016 in Figure 3, was also documented by Plaza et al. (1997) over this area in July 1994, during the final phase of a high O3 period. More recently, Kalabokas et al. (2013, 2015, 2017), Zanis et al. (2014) and Akritidis et al (2016), among others, have shown that similar transport processes of enriched O3 layers at high altitude, can contribute to increase surface O3 concentrations during the summer in the Eastern Mediterranean. This transport has been associated with large scale subsidence within strong northerly winds in the Eastern Mediterranean (Etesian winds), and the affected layers are dryer than average and show negative temperature anomalies. Figure S11 shows the ECMWF ERA-Interim reanalysis together with the AEMET O3 free soundings at Madrid airport for the 13/07/2016. The ridging at the lower troposphere over the Bay of Biscay at the rear of an upper-level trough (left panels) is accompanied by intense NW winds blowing at the middle and upper troposphere and NE winds at ground level and up to 2000 m (see the radiosonde profile in the same figure). The O3 intrusion is associated to the upper-level trough (Sections A-A and B-B in the figure), and a large area of deep subsidence and extremely low relative humidity observed within the NW flows over Madrid and to the north of the Iberian Peninsula and the Bay of Biscay. High O3 concentrations values and low relative humidity of the ERA-Interim profiles over the airport of Madrid (green and red dotted-lines in the panel "g" of the Figure S11) are in agreement with the radiosonde observations in the same panel.

  The question now is how much of this O3 could fumigate at ground level. According to the radiosonde data, the mixing height top was about 2000 m a.s.l. at midday, but could increase to about 3100 m a.s.l. after the projection of the surface temperature increase observed during the afternoon at near-by stations. This height reaches the lower part of the O3 enriched layer originated in the tropopause folding. Thus, a certain impact seems likely. However, the O3 concentrations were relatively low at all surface stations during that day, as it corresponds to a vented low O3 period.

- A new figure S11 has been added and discussed to show the stratospheric intrusion following your suggestion.

- We added these two paragraphs in the conclusions:

- The O3 source apportionment is very complex, having contributions from local/regional and remote sources, including the stratosphere. The relative contributions of these might vary in time and space (e.g. Lefohn et al., 2014).

- ·Climate change might reduce the benefits of the O3 abatement policies (since hot waves increase O3 episodes). This, as well as the measures and policies in Northern America and Asia, will need to be considered into future Europe policies for O3 mitigation (Lefohn and Cooper, 2015; Sicard et al., 2017)

- And we modified the fourth as follows: ·       In the MAB, during the highest O3 (accumulation) episodes, in addition of the contribution (to surface concentrations) by fumigation of upper O3 (from regional transport, hemispheric free troposphere O3, and intruded stratospheric O3, X in Figure 10), there is an added fraction produced locally and transported-recirculated within the MAB, which accumulates from one day to the next (Y in Figure 10).

*Conclusions – Line 587-588: climate change might reduce the benefits of the ozone control strategies. This can be discussed e.g. climate change and the measures and policies in North America or Asia will need to be considered into future ozone policies in Europe for ozone mitigation (Lefohn and Cooper, 2015_Atmo Env; Sicard et al., 2017_ACP).* REPLY. Yes, thanks for this key suggestion for this ¡section on air quality policy. We added 2 bullets:
- ·The O3 source apportionment is very complex, having contributions from local/regional and remote sources, including the stratosphere. The relative contributions of these might vary in time and space (e.g. Lefohn et al., 2014).
- Climate change might reduce the benefits of the O3 abatement policies (since hot waves increase O3 episodes). This, as well as the measures and policies in Northern America and Asia, will need to be considered into future Europe policies for O3 mitigation (Lefohn and Cooper, 2015; Sicard et al., 2017)

*Line 612: a statement of ACP Special Issues (Lamarque et al., 2013_ACP and Young et al. 2013_ACP) with respect to the ACCMIP models can be done, or the validation of ACCMIP ozone simulations using sonde throughout the free troposphere and lower stratosphere for both seasonal and year-to-year variations of ozone (Kazuyuki Miyazaki and Kevin Bowman, 2017_ACP). REPLY. Yes we recognise we may add a bit on these inter-comparison exercises, but the number of tem is large, and come from different scientific-policy forums.* The we decided to insert this text: A good combination of regional/local scale modelling, able to reproduce horizontal/vertical re-circulations of air masses, and the behaviour of urban/industrial plumes in complex topography/meteorology; with modelling able to calculate contributions from long range transport, free troposphere, and stratospheric O3, will be needed to efficiently support policy (see v.g. the ACP special issue on the Atmospheric Chemistry and Climate Model Inter-comparison Project, ACCMIP https://www.atmos-chem-phys.net/special_issue296.html; the FAIRMODE initiative, Thunis et al., 2015; and the Monitoring Atmospheric Composition & Climate (MACC)).

*Grammatical suggestions and typos*
*Lines 69, 88: PM 10 and 2.5 in subscript.* REPLY. Done, thanks
*Line 86: Monks et al., 2015 rather than 2014*. REPLY. Done, thanks
*Line 133: to be reformulated as "recorded in July" (a simpler way).* REPLY. Done, thanks
*Line 142: why did you put (and UFP) with brackets?* REPLY. Brackets deleted

*Line 188: Gómez-Moreno (hyphen)* REPLY. Added, thanks
*Some acronyms need to be defined e.g. lines 206, 215, 236* REPLY. Defined there and some additional ones, thanks
*Line 206: 19 or 20 July, to be checked.* REPLY: It is 19/July, this finished one day before other instruments. Then we leaved it
*Line 442: "through the" (space).* REPLY: Corrected, thanks
*Line 444: remove all the text given in parentheses (really heavy).* REPLY: Deleted, thanks
*Line 475: McKendry and Lundgren (instead of "et al."):* REPLY: Corrected, thanks
*Line 601: add a reference* REPLY: Added: according our calculations (Table S1 and Figure S1).

*Tables & Figures*
*There are too many figures, it would be necessary to select the most informative figures (5-6 maximum) and move the other to SI.* REPLY, as you will see we moved a number of them to SI.
*Please, change "altitude" as "elevation" in Figures and Tables when necessary.* REPLY: Done, thanks for observations and corrections
*Figures 7-10-11 can be joined. Too many Figures seem similar and not useful.* REPLY, We removed Figure 10 and 11 and inserted in the SI.
*Figure 1: put units (m) on X- and Y-axis and "elevation (a.s.l).* REPLY, corrected, many thanks
*Figure 3: how is possible to read variations of values, I don't see the units for each parameter (e.g. T, RH), please add a second Y-axis with units.* REPLY: We added a nex X axis for T as requested
*Figure 4: blurred.* REPLY: We made it again with higher resolution.
*Figure 13: units on Y-axis.* REPLY, done, thanks for the correction
*Table 1: add the station "El Retiro" and define the acronyms in caption (RH, Temp., UFP: : :)* REPLY, done, thanks for the correction
*Table 2: m (a.g.l). :)* REPLY, done, thanks for the correction

*References*
*Kulmala et al., 2000 (Line 702), Millán & Artíñano, 1992 (Line 725) &*
*Skrabalova et al., 2015 are missing.* REPLY, thanks a lot and sorry for these errors. We added cite in the first case and we deleted the 2 last ones.
*Line 766: Pujadas et al., the year is missing (2000).* REPLY. This reference was deleted due to the shortening of text, thank you
*Line 781: "et al.," please supply the full author list. Please, read the advices in the guidelines for authors for the "list of references" (e.g. publication year, lines 787, 702) and consider a chronological order for the same author e.g. Millán.* REPLY, Thanks a lot for detecting these errors. Revised all references.

END OF THE REPORT

---

## Author Response (AR1)

**SUMMARY OF CHANGES DONE FOLLOWING THE REFEREES' SUGGESTIONS**

Dear editor,

Thanks a lot for considering our paper for the revision stage and for the editorial tasks.

As you will see we have implemented the changes and suggestions from both referees. Both reviews were extremely interesting for us, and also coincident in the fact that we miss to better describe the variation of background $O_3$, and the potential stratospheric contributions. You will see that we worked a lot on it and now this is clearly integrated in discussion and interpretation. You will see that we modified a lot the paper in content and format.

Because we requested the support of Dr Milllán to better understand the issue of the $O_3$ subsidence over the Mediterranean, and because he worked to interpret the meteorology to evidence the stratospheric contribution, as well as to prepare the new figure, we would like to introduce him as a co-author of this revised version. I hope you will accept this change also.

Once we finished with revisions we sent the manuscript to Cambridge Proofreading to correct errors in English usage.

We attach the version with track changes in order to easily detect the changes done and another one with these accepted. We also modified the S. I. We also attach a detailed description of how we took into account the referees' comments and suggestions.

We firmly believe that the review process has substantially improved the paper.

Thank you very much again and kind regards

Xavier

Barcelona, Spain, March 5, 2018

**REFEREE #1**

*Overview: The paper deals with the phenomenology of the summer ozone episodes over the greater area of Madrid, Spain. I think that it is a very interesting study, analysing atmospheric measurements of ozone and fine particles together with many other atmospheric parameters and giving further insight to the complicated atmospheric mechanisms related with air pollution over the area. In my opinion, the document deserves publication in ACP, after the recommendations listed below are taken into account.* REPLY, Thanks a lot for considering our manuscript suitable for publication after revision and also for the comments and suggestion on the stratospheric $O_3$ that improved a lot the article.

*General comments: A weak point of the paper is that the levels of measured surface ozone are mainly related (or attributed) to the photochemical ozone production over the metropolitan area of Madrid.* REPLY, Thanks a lot this comment. We also considered long range transport (named 'external' in the paper). But as you might see in the revised version we have enlarged this and also the stratospheric issue that we miss in our original submission.

*On the other hand, I think that the variations of the background ozone levels within the boundary layer and the free troposphere are not discussed with sufficient detail. In relation to that comment and based on research results carried out on the other side of the Mediterranean basin, in the Eastern Mediterranean, it comes out that the regional background ozone levels in the free troposphere and the boundary layer during summer, regularly exceeding 60 ppb, contribute on the average to the greatest part of the surface ozone levels measured in large urban areas like Athens (Kalabokas et al., 2000; Kourtidis et al., 2002; Kouvarakis et al., 2002; Lelieveld et al., 2002; Kalabokas and Repapis, 2004; Gerasopoulos et al., 2005). The main origin of these high ozone background levels over the Eastern Mediterranean is tropospheric ozone subsidence, which seems to be strongly related with specific synoptic meteorological conditions, occurring very frequently during summer at the Eastern side of the Mediterranean basin (Kalabokas et al., 2013; Zanis et al., 2014; Kalabokas et al., 2015; Akritidis et al., 2016). In addition, recent research shows that during springtime ozone episodes (April – May) over the western Mediterranean similar synoptic meteorological patterns might also occur and which are linked with regional episodes mainly induced by large scale tropospheric ozone subsidence, influencing (or fumigating) the boundary layer as well as the ground surface ozone concentrations (Kalabokas et al., 2017).*
REPLY, Thanks a lot for these observations. Yes you are completely right; we miss these issues in our discussions and interpretations. We have added these observations and discussions in several part of the revised paper. For example text added or changed includes:

- Abstract: We added: The results demonstrate the concatenation of venting and accumulation episodes, with relative $O_3$ lows (venting) and peaks (accumulation) in surface levels. Regardless of the episode type, fumigation of high altitude $O_3$ (from different origins) contributes the major proportion of surface $O_3$ concentrations.
- Section 1: New text added: In the Eastern Mediterranean, the regional background O3 levels in the free troposphere and the boundary layer during summer might regularly exceed 60 ppb, and fumigation of these upper air masses contribute on the average to the greatest part of the surface O3 levels measured in Greece (Kalabokas et al., 2000; Kourtidis et al., 2002; Kouvarakis et al., 2002; Lelieveld et al., 2002; Kalabokas and Repapis, 2004; Gerasopoulos et al., 2005). Furthermore, a number of studies reported contributions of stratosphere to the surface O3 concentrations during specific meteorological scenarios in the same region (Kalabokas et al., 2013, 2015; Zanis et al., 2014; Parrish et al., 2012; Lefohn et al. , 2012; Akritidis et al., 2016, among others). In addition, recent research shows that during springtime O3 episodes (April – May) over the WMB similar synoptic meteorological patterns might also occur; and that these are linked with regional episodes mainly induced by large scale tropospheric O3 subsidence, influencing the boundary layer as well as the ground surface O3 concentrations (Kalabokas et al., 2017). However, the most intense episodes in the WMB occur in June-July according to the statistics for the 2000-2015 period by Querol et al. (2016) for Spain.

- Section 3.1. New text added: The AEMET free-sounding shows low O3 surface concentrations (<45 ppb) and high levels (>70 ppb) in the middle troposphere (3000-5000 m a.s.l.), associated with very low relative humidity and intense W to NW winds blowing at that height, which will be discussed in section 4.
- Section 4: discussion added, see below.
- New figure added (see below)
- Conclusions modified (see below)

*Even if the typical meteorological conditions prevailing over the Iberian Peninsula during summer are quite different than in the Eastern Mediterranean, as it is very well described in the introduction of the manuscript, occasionally such conditions might occur. In fact, I think that this is the case of the ozone episode of 11-15 July 2016, which is the most studied period in the manuscript (when the intensive measuring campaign has taken place). As shown in Figs 3 and 12, the free tropospheric ozone levels are much higher on July 13, 2016 than the two weeks before and after and at the same time the relative humidity values in the lower troposphere are close to zero (and being in sharp contrast with the periods before and after). In fact, this feature is a very common characteristic of deep and large-scale tropospheric subsidence in summertime ozone vertical profiles over the Eastern Mediterranean, indicating an origin of air masses from the upper tropospheric or stratospheric layers (Kalabokas et al., 2013; Kalabokas et al., 2015).*

*Therefore, for a better assessment of the free tropospheric influence as well as the reported fumigation events over the area, I would suggest putting more emphasis on the analysis of the synoptic conditions during this most studied period, when the intensive measurement campaign has taken place (11-15 July 2016). A figure could be added including at least the daily meteorological maps of geopotential height, omega vertical velocity and specific humidity at 700hPa pressure level (representative for the free troposphere), which I think that they would be sufficient to follow satisfactorily the evolution and the geographical extent of the subsidence phenomenon (the subsiding air mass seems to originate from N-NW Atlantic). If this parameter is taken into account, then I think that the discussion concerning the origin of the fumigation events during 11-16 July 2016 would be more complete (tropospheric ozone subsidence in addition to the local ozone photochemical production associated with valley-breeze recirculation and ozone residual layers, as mentioned many times in the manuscript). So, I would suggest modifying accordingly the respective paragraphs, where sometimes the high ozone values recorded at the top of the boundary layer are not fully explained (e.g.: Page 5, lines 175-187; Page 9, lines 348 – 352, 362-365; Page 12, lines 470 – 473, 483-487; Page 14, lines 570 – 582).*

REPLY. Thanks a lot for these very interesting comments. Indeed we have applied all your suggestions and modified the introduction, discussion and conclusions sections to this end. Examples are:

- Section 4. We added the following discussion: Considering the free sounding O3 profiles in Figure 3, high O3 concentrations (>70 ppb) can be observed above the PBL, between 3000 and 5000 m a.s.l., which may be related to larger scale transport of pollutants, previously uplifted to the mid-troposphere or originated after a stratospheric intrusion and a subsequent deep subsidence into the middle troposphere, as it is probably the case based on the ECMWF ERA-Interim reanalysis data. Transport of high O3 air masses in the middle troposphere, as for the 13/07/2016 in Figure 3, was also documented by Plaza et al. (1997) over this area in July 1994, during the final phase of a high O3 period. More recently, Kalabokas et al. (2013, 2015, 2017), Zanis et al. (2014) and Akritidis et al (2016), among others, have shown that similar transport processes of enriched O3 layers at high altitude, can contribute to increase surface O3 concentrations during the summer in the Eastern Mediterranean. This transport has been associated with large scale subsidence within strong northerly winds in the Eastern Mediterranean (Etesian winds), and the affected layers are dryer than average and show negative temperature anomalies. Figure S11 shows the ECMWF ERA-Interim reanalysis together with the AEMET O3 free soundings at Madrid airport for the 13/07/2016. The ridging at the lower troposphere over the Bay of Biscay at the rear of an upper-level trough (left panels) is accompanied by intense NW winds blowing at the middle and upper troposphere and NE winds at ground level and up to 2000 m (see the radiosonde profile in the same figure). The O3 intrusion is associated to the upper-level trough (Sections A-A and B-B in the figure), and a large area of deep subsidence and extremely low relative humidity observed within the NW flows over Madrid and to the north of the Iberian Peninsula and the Bay of Biscay. High O3 concentrations values and low relative humidity of the ERA-Interim profiles over the airport of Madrid (green and red dotted-lines in the panel "g" of the Figure S11) are in agreement with the radiosonde observations in the same panel.

The question now is how much of this O3 could fumigate at ground level. According to the radiosonde data, the mixing height top was about 2000 m a.s.l. at midday, but could increase to about 3100 m a.s.l. after the projection of the surface temperature increase observed during the afternoon at near-by stations. This height reaches the lower part of the O3 enriched layer originated in the tropopause folding. Thus, a certain impact seems likely. However, the O3 concentrations were relatively low at all surface stations during that day, as it corresponds to a vented low O3 period.

- A new figure S11 has been added and discussed to show the stratospheric intrusion following your suggestion.
- We added these two paragraphs in the conclusions:
- The O3 source apportionment is very complex, having contributions from local/regional and remote sources, including the stratosphere. The relative contributions of these might vary in time and space (e.g. Lefohn et al., 2014).

- Climate change my reduce the benefits of the $O_3$ abatement policy (since hot waves increase $O_3$ episodes), and this, as well as the measures and policies in N America and Asia will need to be considered into future Europe policies for $O_3$ mitigation (Lefohn and Cooper, 2015; Sicard et al., 2017).

- And we modified the fourth as follows: ·        ·In the MAB, during the highest O3 (accumulation) episodes, in addition of the contribution (to surface concentrations) by fumigation of upper O3 (from regional transport, hemispheric free troposphere O3, and intruded stratospheric O3, X in Figure 10), there is an added fraction produced locally and transported-recirculated within the MAB, which accumulates from one day to the next (Y in Figure 10).

*In addition, I think that it would be more appropriate to refer to "Western Mediterranean" ozone when analyzing ozone in Spain (instead of "Mediterranean" or "S. Europe" in general) as, according to the above mentioned papers, the phenomenology of the summertime ozone over the Eastern Mediterranean seems to be is quite different than the typical ozone phenomenology over the Western Mediterranean.*

REPLY: Yes we have revised and changed following your suggestion. Thanks for this.

*I would strongly recommend taking these considerations into account, which have been made in the spirit to further improve this good quality manuscript, by modifying the respective paragraphs. After responding to these remarks, I think that the paper is ready for publication.*

REPLY: Yes we did implement your very good observations and we thank you a lot for your very valuable review.

*Technical comments: Fig. 4: Very condensed and difficult to follow, especially on printed paper. I would suggest splitting into two parts and eventually using gridlines.*

REPLY: We made it again with clearer drawings.

END OF THE REPORT

**REFEREE #2**

**Comments from referee**
*The study aimed to explore if changes in nucleation of ultrafine particles and high Surface ozone levels in urban areas could be related to the atmospheric dynamics changes over the European region at highest ozone risk. The content and the methodology are consistent and this study reports valuable conclusions based on experimental investigation by means of simultaneous vertical profiles of concentrations for both pollutants and meteorological parameters using balloons.*
*However, the conclusions can be improved (other than ozone levels are low with thick PBL and high wind speed) and some parts are really heavy going and need to be reformulated. In general, the English language can be improved and some parts need to be reformulated in a simpler way. The entire paper needs to be (deeply) reorganized (i.e. section Results & Discussions & Materials and Methods) and shortened.*
REPLY, Thanks a lot for considering our manuscript suitable for publication after revision and also for the detailed revision done that improved a lot the presentation of our results. We have also revised again English usage.

*The term "altitude" must be correctly used, i.e. use "altitude" for a.g.l. (for vertical profiles) and "elevation" for a.s.l. Please, revise all figures and manuscript.* REPLY: Corrected in all text, figures and tables.

*Abstract - Line 34: "high O3 level" or "ground-level O3".* REPLY: Done.

*Introduction - The state-of-the-art is consistent, however for a non-European scientist; the references are limited to Spain instead of Western Mediterranean region. The state-of-the-art must be documented e.g. including an overview of ozone impacts (Ochoa-Hueso et al., 2017, EnPo) and I found different studies (Lelieveld et al., 2002_Science; Kalabokas et al., 2008_ Atm Env; Giannakopoulos et al., 2009_Global&Planetary Change; Velchev et al., 2011_ACP; Sicard et al., 2013_Atm Env).*
REPLY: Thanks for the constructive comment. Yes indeed we went directly to focus referencing of Millan's team in Spain, because they were pioneering the ozone research in this region in the 1980s-1990s, and in our opinion the built the basis of knowledge for atmospheric dynamics governing O3 levels in The Western Mediterranean, clearly different from those prevailing in the Eastern side of the basin. We recognize now that we missed first reviewing the whole Mediterranean issue. We have added now this view and included references you supplied us with and commented them in the introduction and discussion sections.

*Particulate matter and tropospheric ozone are the most threatening air pollutants in cities (EEA, 2015). More than 75% of the urban population is exposed to levels exceeding WHO guidelines for PM2.5, PM10 an surface O3 (EEA, 2015).* REPLY, we understand that you require us to add this into our text, and we did it using updated EEA (2017). Thanks

*Line 66: "implemented" by?* REPLY: Added: by the national, regional and local administrations

*Line 70: add a reference.* REPLY Added: EEA, 2017

*Line 73: add an example for "easy to identify".* REPLY: Added: (such as abating industrial, shipping and traffic emission with catalytic converters for $NO_X$ and particulate controls for PM).

*Lines 76-77, add a reference, what are the climatic and geographical characteristics leading to high ozone levels? In summer, the western part of the Mediterranean basin is dominated by anti-cyclonic subsidence (Hadley circulation +Azores), high pressure, low winds and strong insolation and thus atmospheric stability favouring photochemical production of $O_3$ (Kalabokas et al., 2008; Giannakopoulos et al., 2009; Velchev et al., 2011; Sicard et al., 2013) and emissions of biogenic VOCs (Giannakopoulos et al., 2009).* REPLY: Added: In summer, the Western Mediterranean Basin (WMB), surrounded by high mountains, falls under the influence of the semi-permanent Azores anticyclone. Clear skies prevail under a generalized level of subsidence aloft, and meso-meteorological processes with marked diurnal cycles dominate. Re-circulations, strong insolation and stability of upper layers favour the production/accumulation of O3 (Millán et al., 1997, 2000 and 2002; Kalabokas et al., 2008; Giannakopoulos et al., 2009; Velchev et al., 2011; Sicard et al., 2013), and the emissions of biogenic volatile organic compounds (BVOCs) (Giannakopoulos et al., 2009).

*Line 90: add a value and a reference for "peaks O3 concentrations".* REPLY: We added (those exceeding the hourly information threshold of 180 $\mu g/m^3$)

The ozone control measures are effective at rural sites while ozone levels are rising in cities (Paoletti et al., 2014_EnPo; Sicard et al., 2013_Atm Env). REPLY: Thank you, yes we added these references to the others that find this trend in Europe. We missed them and these are very important.

*Line 99: NO titration, add a reference. What is the situation in Spain as compared to other European countries such as France and Italy?* REPLY: Yes we added reference; This NO/NO2 is widely accepted in Europe. We added: This trend to decrease the NO/NO2 rate from diesel vehicle emissions (the main source of NOx in urban Europe) has been widely reported (i.e. Carslaw et al., 2016)

*Line 107: the urban areas are characterized by "VOC-limited" conditions, and a reduction in NOx emission increases the O3 formation.* REPLY, Yes we also gave this possibility in the original paper, but to make it more clear we added: We still do not know if this increase is due to a decrease in the NO titration effect or to the fact that the O3 formation is by VOCs dominated, since the urban areas are characterized by 'VOC-limited' conditions, and a reduction in NOx emission might yield to an increase of the O3 formation.

*Lines 112-116: too many references and autocitation, the state-of-the-art need to be enlarged (e.g. Italy, France, Portugal) and add a comparison + quantification of the spatio-temporal changes for surface ozone levels.* REPLY. OK we changed this focus on the whole Mediterranean and we added the references (but no auto-citation, no one paper of the main author of this paper is cited here). Now we stated: Intensive research on O3 pollution has been carried out in the Mediterranean since the late 1980s, which has been key to understand the behaviour of this pollutant in Europe, and to establish the current air quality European standards (Millán et al., 1991, 1996a, 1996b, 1996c, 2000, 2002; Millán, 2002; Lelieveld 2002; EC, 2002, 2004; Millán and

Sanz, 1999; Mantilla et al., 1997; Salvador et al., 1997, 1999; Gangoiti et al., 2001; Stein et al., 2004, 2005; Chevalier et al. , 2007; Kalabokas et al., 2008, 2015 and 2017; Castell et al., 2008a, 2008b, 2012; Kulkarni et al., 2011; Velchev et al., 2011; Doval et al., 2012; Sicard et al., 2013; Millán et al., 2014; Escudero et al., 2014; Zanis et al., 2014; Sicard et al., 2017; among others). EEA (2017) reports a clear gradient to increase the exceedances of the human protection 8-h O3 target value in Southern and Central Europe, higher in the Italian Po Valley and Spain, and relatively lower in Portugal and the Eastern Mediterranean.

*Lines 124-127: to be reformulated.* REPLY: Done: Querol et al. (2017) reported that the high-O3 plume transported from the metropolitan area of Barcelona contributed decisively to cause the frequent exceedances of the information threshold in the northern areas of Barcelona during the acute O3 episodes in July 2015.

*Line 128: what are the scenarios?* REPLY: We modified the text: They also demonstrated that the meteorology associated was very complex, similar to the scenarios of vertical recirculation of air masses reported by………….

*Line 139: "lowpre-existing: : :", please explain. REPLY: Modified to: ….* low condensation sink potential (i.e. relatively clean atmospheres with low surface aerosol concentrations).

*Line 142: add the period of the study.* REPLY: Added.

*Materials & methods - The experimental protocol and methodology are consistent, well described. Even if this section is important, it can be shortened, e.g. lines 165-190: move to section "Discussion" and need to be shortened.* REPLY: Done. We still believe that the right place was the first description of the study area, but following requirements of the referee we moved to 'discussion' and shortened.

*Line 195: "highest levels of O3", how much?* REPLY: Added (with hourly maxima sporadically exceeding 180 µg/m$^3$)

*Line 218: "very good results", please quantify (e.g. r2) this statement.* REPLY: Added to text (with R$^2$ reaching 0.65-0.98 and slopes 0.87-1.23, Minguillón et al., 2015).
*Results – Please avoid describing Figures (e.g. 260-262, 318-320), this is really heavy*. REPLY: Deleted in both cases.
*Lines 249-253: move to section "Materials & Methods".* Reply: Done

*Line 254: what is the correlation coefficient between NO2 and ozone?* REPLY, below you have it for 2 of the sites having the highest O3 (NO2 is in Y, and O3 in X),

[Figure]

*Lines 271-272: move to section "Discussion".* REPLY: We do not understand here what you mean, moving 2 lines breaks the explanation of this paragraph. We decided to keep them in.
*Lines 260-264: move to section "Materials & Methods".* REPLY: In a prior observation you suggested deleting this description and we did it. Now is only 2 lines well inserted in this section.

*Line 344: "high hourly O3 concentrations: ::"* REPLY: Corrected and added, thanks
*Lines 345-348: move to section "Discussion".* REPLY: Moved

*Lines 353-414: long section, boring, need to be shortened.* REPLY We reduced it by 60% by moving descriptions of the first 3 days of soundings to SI and synthesising the results in the main text.

*Line 441: "biogenic VOCs", please specify the source (e.g. isoprene).* REPLY: Added: The influence of aromatic VOCs on OFP rapidly decreases while the influence of biogenic VOCs (mostly isoprene followed by monoterpenes, as primary ones, and methacrolein, methyl-vinyl-ketone, isoprene-derived isomers of unsaturated hydroxy hydroperoxides (ISOPOOH) and methylglyoxal, as the main secondary species)…

Lines 459-464: move to section "Results". REPLY. This is one of the only two cases that we decided to keep the original text here because, this discussion is summarising the evolution of the UFP and O3 profiles across the day, If we move these 5 lines, we start description from midday here and we lose the early to midday scenarios.

*Line 468: titration by NO.* REPLY: Added, thanks

*Lines 487-495: to be reformulated, unclear.* REPLY: Re-written: During the whole month of July 2016 there was a clear veering of the urban plume from Madrid, with night plume transport towards SW (MJDH-San Martin de V., Figures 9 and S12), and towards NW, N-NE, and, in some cases, E-SE during the morning and midday, followed by the decoupling and onset of the evening and nocturnal flow towards SW. This veering seems to be causally associated with the high O3 levels recorded in the W to E areas surrounding northern Madrid, since the peak concentrations recorded by the official air quality network follow this spatial and temporal evolution (Figure S12) for the exceedances of the O3 information threshold.

*Lines 500-505: typical diurnal ozone concentrations, well documented in the literature, not innovative for ACP.* REPLY: Done, we delete it.

*Lines 506-5014: move to section "Results".* REPLY. This is one of the two cases that we decided to keep the original text here because, this discussion is summarising the evolution of the UFP and O3 profiles across the day.

*Lines 518-519: confusing compared to line 560, need to be reformulated.* *REPLY, thanks a lot, changed to:* Conversely, the lower development of the PBL on 14/07/2016 causing les surface UPF dilution and lower top-down contributions to $O_3$ to surface concentrations accounted for opposite $O_3$ and UFP the profiles.

*Lines 538-541: move to section "Results". At high elevation, changes in the background tropospheric ozone can be attributed to to i) hemispheric background concentrations, in part due to the reduction in NOx emissions; ii) the exchange between the free troposphere and the boundary layer and iii) the stratospheric inputs (Chevalier et al., 2007_ACP, Kulkarni et al., 2011; Lefohn et al., 2012_Atm Env; Sicard et al., 2016_Env Research). The "stratospheric inputs" need to be discussed in this section, as well as exchanged between the lower stratosphere and free troposphere.* REPLY: Yes this was a weakness, thanks a lot. We added: In addition of the local O3, the background contribution can be also very relevant. At high elevation, changes in the background tropospheric O3 can be attributed to to (i) hemispheric background concentrations; (ii) exchange between the free troposphere and the boundary layer; and iii) stratospheric inputs (Chevalier et al., 2007; Kulkarni et al., 2011; Parrish et al., 2012, Lefohn et al., 2012; Kalabokas et al., 2013, 2015 and 2017; Zanis et al., 2014; Akritidis et al., 2016; Sicard et al., 2017).
We also modified several parts to discuss these free-troposphere and stratospheric contributions. Examples are:

- Section 4. We added the following discussion: Considering the free sounding O3 profiles in Figure 3, high O3 concentrations (>70 ppb) can be observed above the PBL, between 3000 and 5000 m a.s.l., which may be related to larger scale transport of pollutants, previously uplifted to the mid-troposphere or originated after a stratospheric intrusion and a subsequent deep subsidence into the middle troposphere, as it is probably the case based on the ECMWF ERA-Interim reanalysis data. Transport of high O3 air masses in the middle troposphere, as for the 13/07/2016 in Figure 3, was also documented by Plaza et al. (1997) over this area in July 1994, during the final phase of a high O3 period. More recently, Kalabokas et al. (2013, 2015, 2017), Zanis et al. (2014) and Akritidis et al (2016), among others, have shown that similar transport processes of enriched O3 layers at high altitude, can contribute to increase surface O3 concentrations during the summer in the Eastern Mediterranean. This transport has been associated with large scale subsidence within strong northerly winds in the Eastern Mediterranean (Etesian winds), and the affected layers are dryer than average and show negative temperature anomalies. Figure S11 shows the ECMWF ERA-Interim reanalysis together with the AEMET O3 free soundings at Madrid airport for the 13/07/2016. The ridging at the lower troposphere over the Bay of Biscay at the rear of an upper-level trough (left panels) is accompanied by intense NW winds blowing at the middle and upper troposphere and NE winds at ground level and up to 2000 m (see the radiosonde profile in the same figure). The O3 intrusion is associated to the upper-level trough (Sections A-A and B-B in the figure), and a large area of deep subsidence and extremely low relative humidity observed within the NW flows over Madrid and to the north of the Iberian Peninsula and the Bay of Biscay. High O3 concentrations values and low relative humidity of the ERA-Interim profiles over the airport of Madrid (green and red dotted-lines in the panel "g" of the Figure S11) are in agreement with the radiosonde observations in the same panel.

The question now is how much of this O3 could fumigate at ground level. According to the radiosonde data, the mixing height top was about 2000 m a.s.l. at midday, but could increase to about 3100 m a.s.l. after the projection of the surface temperature increase observed during the afternoon at near-by stations. This height reaches the lower part of the O3 enriched layer originated in the tropopause folding. Thus, a certain impact seems likely. However, the O3 concentrations were relatively low at all surface stations during that day, as it corresponds to a vented low O3 period.

- A new figure S11 has been added and discussed to show the stratospheric intrusion following your suggestion.
- We added these two paragraphs in the conclusions:
- The O3 source apportionment is very complex, having contributions from local/regional and remote sources, including the stratosphere. The relative contributions of these might vary in time and space (e.g. Lefohn et al., 2014).

- ·Climate change might reduce the benefits of the O3 abatement policies (since hot waves increase O3 episodes). This, as well as the measures and policies in Northern America and Asia, will need to be considered into future Europe policies for O3 mitigation (Lefohn and Cooper, 2015; Sicard et al., 2017)

- And we modified the fourth as follows: ·        In the MAB, during the highest O3 (accumulation) episodes, in addition of the contribution (to surface concentrations) by fumigation of upper O3 (from regional transport, hemispheric free troposphere O3, and intruded stratospheric O3, X in Figure 10), there is an added fraction produced locally and transported-recirculated within the MAB, which accumulates from one day to the next (Y in Figure 10).

*Conclusions – Line 587-588: climate change might reduce the benefits of the ozone control strategies. This can be discussed e.g. climate change and the measures and policies in North America or Asia will need to be considered into future ozone policies in Europe for ozone mitigation (Lefohn and Cooper, 2015_Atmo Env; Sicard et al., 2017_ACP).* REPLY. Yes, thanks for this key suggestion for this ¡section on air quality policy. We added 2 bullets:

- ·The O3 source apportionment is very complex, having contributions from local/regional and remote sources, including the stratosphere. The relative contributions of these might vary in time and space (e.g. Lefohn et al., 2014).
- Climate change might reduce the benefits of the O3 abatement policies (since hot waves increase O3 episodes). This, as well as the measures and policies in Northern America and Asia, will need to be considered into future Europe policies for O3 mitigation (Lefohn and Cooper, 2015; Sicard et al., 2017)

*Line 612: a statement of ACP Special Issues (Lamarque et al., 2013_ACP and Young et al. 2013_ACP) with respect to the ACCMIP models can be done, or the validation of ACCMIP ozone simulations using sonde throughout the free troposphere and lower stratosphere for both seasonal and year-to-year variations of ozone (Kazuyuki Miyazaki and Kevin Bowman, 2017_ACP). REPLY. Yes we recognise we may add a bit on these inter-comparison exercises, but the number of tem is large, and come from different scientific-policy forums.* The we decided to insert this text: A good combination of regional/local scale modelling, able to reproduce horizontal/vertical re-circulations of air masses, and the behaviour of urban/industrial plumes in complex topography/meteorology; with modelling able to calculate contributions from long range transport, free troposphere, and stratospheric O3, will be needed to efficiently support policy (see v.g. the ACP special issue on the Atmospheric Chemistry and Climate Model Inter-comparison Project, ACCMIP https://www.atmos-chem-phys.net/special_issue296.html; the FAIRMODE initiative, Thunis et al., 2015; and the Monitoring Atmospheric Composition & Climate (MACC)).

*Grammatical suggestions and typos*
*Lines 69, 88: PM 10 and 2.5 in subscript.* REPLY. Done, thanks
*Line 86: Monks et al., 2015 rather than 2014.* REPLY. Done, thanks
*Line 133: to be reformulated as "recorded in July" (a simpler way).* REPLY. Done, thanks
*Line 142: why did you put (and UFP) with brackets?* REPLY. Brackets deleted
*Line 188: Gómez-Moreno (hyphen)* REPLY. Added, thanks
*Some acronyms need to be defined e.g. lines 206, 215, 236* REPLY. Defined there and some additional ones, thanks
*Line 206: 19 or 20 July, to be checked.* REPLY: It is 19/July, this finished one day before other instruments. Then we leaved it
*Line 442: "through the" (space).* REPLY: Corrected, thanks
*Line 444: remove all the text given in parentheses (really heavy).* REPLY: Deleted, thanks
*Line 475: McKendry and Lundgren (instead of "et al."):* REPLY: Corrected, thanks
*Line 601: add a reference* REPLY: Added: according our calculations (Table S1 and Figure S1).

*Tables & Figures*
*There are too many figures, it would be necessary to select the most informative figures (5-6 maximum) and move the other to SI. REPLY, as you will see we moved a number of them to SI.*
*Please, change "altitude" as "elevation" in Figures and Tables when necessary.* REPLY: Done, thanks for observations and corrections
*Figures 7-10-11 can be joined. Too many Figures seem similar and not useful.* REPLY, We removed Figure 10 and 11 and inserted in the SI.
*Figure 1: put units (m) on X- and Y-axis and "elevation (a.s.l).* REPLY, corrected, many thanks
*Figure 3: how is possible to read variations of values, I don't see the units for each parameter (e.g. T, RH), please add a second Y-axis with units.* REPLY: We added a nex X axis for T as requested
*Figure 4: blurred. REPLY: We made it again with higher resolution.*
*Figure 13: units on Y-axis.* REPLY, done, thanks for the correction
*Table 1: add the station "El Retiro" and define the acronyms in caption (RH, Temp., UFP: : :)* REPLY, done, thanks for the correction
*Table 2: m (a.g.l). :)* REPLY, done, thanks for the correction

- The phenomenology of $O_3$ episodes in S Europethe WMBestern Mediterranean is extremely complex, mainly due to the close couplingrelation between photochemistry processes and mesoscale atmospheric dynamics. This , requiresing, consequently, abatement policies very different to from the ones useful for Central Northern Europe, as intensive research has demonstrated in the last decades.

- In the MAB, dDuring the highest $O_3$ (accumulation) episodes, to in addition to the apart from the contribution (to surface concentrations) of remote sources, afterby fumigation of upper $O_3$ enriched layers fumigation(from regional transport, hemispheric free troposphere $O_3$, and intruded stratospheric $O_3$) ,contribution ( X in Figure 10)5) to surface $O_3$ concentrations, there is an added fraction of $O_3$ produced locally andor transported recirculated horizontallywithin the MAB, which accumulates from one day to the next (Y in Figure 105). If sensitivity analyses demonstrate that abatement of specific precursors would have an effect on reducing $O_3$ peaks, then the reduction strategies (geographic extension, timing, and so on...) to for decrease decreasing the YX and XY components are very different, and, in most cases, the X component will dominate the relative contributions. Thus, probably, structural measures over wider regions would be more effective than local episodic measures, ( which might have a larger effect on the Y component). In terms of precursors, the OFP analysis carried out at the…)  ISCIII site shows that, even if anthropogenic emissions may  dominate the $O_3$ formation through the potential impact of alkenes and alkanes (not measured ), and the high contribution of carbonyls (formaldehyde and acetaldehyde), biogenic emissions must be considered. Biogenic VOC (primary and secondary) and aromatic compounds (C6 to C10) contribute to the same extent to the OFP, according our calculations (Table S1 and Figure S1).

- The meteorological scenarios causing the summer accumulation episodes in the MAB (high temperatures, low synoptic winds and relatively thinner PBL) should be forecast in order, to drive an effective alert system .

- A more detailed characterisation of $O_3$ precursors (VOC and  BVOCs) in the MAB is necessary, especially in the source areas, to effectively predict the photochemical evolution of the plumes,  the main impact areas where $O_3$ from high--altitude reservoir layers formed the previous day(s)  fumigates to the surface levels enriched in $O_3$ and other precursors.

- Modelling techniques and sensitivity analyses  will allow  the  simulation  of  real conditions  concerning  $O_3$ abatement potential only if the following is achieved in advance: i)  the recirculation cells and other local/regional  meteorological p as the fumigation timing  and regional  plume transport, are reproduced; ii)  a geographically resolved and accurate emission inventory of $O_3$ precursors in the source areas and their temporal modulation is included; and iii) he origin of the high--altitude $O_3$ strata from external origins is reproduced.

- A good combination of regional/local scale modelling, able to reproduce horizontal/vertical recirculations of air masses and the behaviour of  urban/industrial plumes in complex topography/meteorology, with modelling  able to calculate contributions from long-range transport, free troposphere, and stratospheric $O_3$, will be needed to efficiently support policy (see, for example, the ACP special issue on the Atmospheric Chemistry and Climate Model Intercomparison Project, ACCMIP https://www.atmos-chem-phys.net/special_issue296.html; the FAIRMODE initiative, Thunis et al., 2015; and the Monitoring Atmospheric Composition & Climate (MACC)).

The conceptual model described  in this study for $O_3$ episodes in the MMA confirm the relevance of the vertical re-circulations (on top of the high atmospheric multi-source $O_3$ background) that Millan et al (1997, 2000), Gangoiti et al. (2001), and Millán (2014)

highlighted, and controlled in this case by specific synoptic conditions and the PBL depth, may
also be also applicable to most of the S EuropeWestern Mediterranean Basin (WMB). Thus,

[revised manuscript text omitted]